# Continuous speech with pauses inserted between words increases cortical tracking of speech envelope

Suwijak Deoisres[1]*, Yuhan Lu[2], Frederique J. Vanheusden[3], Steven L. Bell[1], David M. Simpson[1]

1 Institute of Sound and Vibration Research, University of Southampton, Southampton, United Kingdom,
2 Key Laboratory for Biomedical Engineering of Ministry of Education, College of Biomedical Engineering and Instrument Sciences, Zhejiang University, Hangzhou, China, 3 Department of Engineering, School of Science and Technology, Nottingham Trent University, Nottingham, United Kingdom

* sd1n17@soton.ac.uk

**Data Availability Statement:** All EEG and stimulus files are now available from the University of

## Abstract

The decoding multivariate Temporal Response Function (decoder) or speech envelope reconstruction approach is a well-known tool for assessing the cortical tracking of speech envelope. It is used to analyse the correlation between the speech stimulus and the neural response. It is known that auditory late responses are enhanced with longer gaps between stimuli, but it is not clear if this applies to the decoder, and whether the addition of gaps/pauses in continuous speech could be used to increase the envelope reconstruction accuracy. We investigated this in normal hearing participants who listened to continuous speech with no added pauses (natural speech), and then with short (250 ms) or long (500 ms) silent pauses inserted between each word. The total duration for continuous speech stimulus with no, short, and long pauses were approximately, 10 minutes, 16 minutes, and 21 minutes, respectively. EEG and speech envelope were simultaneously acquired and then filtered into delta (1–4 Hz) and theta (4–8 Hz) frequency bands. In addition to analysing responses to the whole speech envelope, speech envelope was also segmented to focus response analysis on onset and non-onset regions of speech separately. Our results show that continuous speech with additional pauses inserted between words significantly increases the speech envelope reconstruction correlations compared to using natural speech, in both the delta and theta frequency bands. It also appears that these increase in speech envelope reconstruction are dominated by the onset regions in the speech envelope. Introducing pauses in speech stimuli has potential clinical benefit for increasing auditory evoked response detectability, though with the disadvantage of speech sounding less natural. The strong effect of pauses and onsets on the decoder should be considered when comparing results from different speech corpora. Whether the increased cortical response, when longer pauses are introduced, reflect improved intelligibility requires further investigation.

Southampton eprints repository (dataset DOI: doi:10.5258/SOTON/D2281).

**Funding:** The authors received no specific funding for this work.

**Competing interests:** The authors have declared that no competing interests exist.

## Introduction

Auditory evoked responses (AERs) represent brain activity following auditory stimulation that may be transient, such as clicks, tones or phonemes, or continuous, such as modulated tones or speech. Interest in continuous speech responses has increased greatly in recent years, with the desire to use ecologically more valid stimuli that reflect real-world listening challenges, which has been accompanied by the development of new analysis tools for natural stimuli. Responses can be recorded invasively, e.g. through the electrocorticogram (ECoG) [1], or non-invasively e.g., through the magneto- or electro- encephalogram (MEG/EEG) [2, 3]. AERs to repeating short stimuli are well established in assessing hearing impairment and some neurological disorders. However, hearing loss does not impact only the ability to detect sounds (as measured in a standard pure tone audiogram), but also the intelligibility of speech, especially in the presence of other sounds such as background noise or competing speech. Using natural speech stimuli to measure responses provides a benefit over using transient stimuli in that hearing speech is the primarily concern of hearing impaired people and so testing real-world listening challenges with speech has ecological validity [4].

It is widely observed that human brain activity shows tracking of the envelope of stimulus when listening to natural speech [5, 6]. One of the hypothesised functional roles of cortical tracking of speech envelope is that it represents the tracking of acoustic onsets [2, 6]. The acoustic onsets in natural speech are generally most commonly linked to the syllable boundaries and regions where speech sounds occur after silent pauses, but it remains unclear how acoustic onsets contribute to cortical envelope tracking. For conventional AERs to repeating stimuli, particularly the auditory late response (ALR), the acoustic onsets can be clearly identified at the start of each sound and it is well established that longer intervals (silent gaps) between stimuli enhances the onset response [7]. A clear observation of strong effects of onsets on cortical responses to natural speech was reported by Hamilton, Edwards [1] using invasive ECoG measurement, in which some regions of the Superior Temporal Gyrus were very sensitive to onset portions of speech, whilst other regions of the Gyrus appeared sensitive to more sustained speech components after onsets. However, it is not clear to what extent cortical tracking reflects onset or sustained responses to speech, and hence if we should expect such cortical tracking to increase with the addition of gaps or pauses in continuous speech.

Speech rate is one of the elements which influences individual's speech perception [8–10], which can be associated with pauses in the speech stream. In audiological research, this element is often manipulated by compressing or expanding the temporal waveform of the speech stimulus [8–11]. The effect of compressed speech (faster speech rate) and expanded speech (slower speech rate) on intelligibility is highly variable. Compressed speech and expanded speech have both been found to increase [8, 11] and decrease [9, 10, 12] individual's speech intelligibility and in some cases to cause no effect [13]. Time-compressed and -expanded speech both cause distortion in the acoustic signal. It is clear that they have a strong effect on intelligibility when compressed or expanded more than 0.5 times relative to the original speech rate [10, 14]. It is currently thought that that intelligibility of time-compressed and -expanded speech relates to the individual's cognitive processing capabilities [10]. It has been suggested that changes in brain function with age can influence speech intelligibility [15–17]. It has been reported that older adults, especially those who have reduced cognitive function, benefit from listening to speech presented at a slower rate, as there is more time to process linguistic information [18, 19]. A challenge in interpreting responses to compressed and expanded speech is that it is difficult to disentangle the effects of changes in speech intelligibility from various acoustic properties of the stimuli, for example, intensity envelope and duration of pauses [11, 20, 21].

A few EEG studies investigated the effect of pauses in speech to the cortical response to continuous speech. First, Kayser, Ince [22] investigated the effect of an irregular speech rate on auditory responses in different brain locations and different frequency bands of the EEG. The modification of speech stimuli was carried out by extending or shrinking existing pauses between syllables and words randomly with limits to the modified pause of not more than three times the original duration. Auditory responses to speech with irregular rate generated weaker left frontal alpha power and cortical tracking in the delta frequency band. The weaker responses were suggested to be related to reduce top-down control, e.g., less attention to stimuli, by the frontal and premotor cortices over the auditory cortex. Another study by Hambrook, Soni [23] modified speech stimuli by inserting periodic silent pauses to replace speech sounds. They consistently found weaker cortical tracking of speech. It is suggested that the interruption of silent pauses in speech degrades the acoustic properties and the rhythm. Some studies manipulated pauses in speech (typically pauses between phrases and sentences) by shortening them to 0.3–0.5 seconds. For example, in dual attention task studies by Power, Foxe [24] and Kong, Mullangi [25], long pauses were truncate to minimise subject's attention to the speech they were informed to ignore when there is silence in the target speech, and to make the speech stimulus more continuous.

To the best of the authors knowledge, no study involving AERs utilized the method of inserting fixed duration pauses between phrases into speech, a method so far only used in behavioural tests such as studies by Tanaka, Sakamoto [19] and Ghitza and Greenberg [21]. Therefore, in this study we investigate how continuous speech with additional fixed pauses inserted between words affects human cortical auditory responses. The aim is to further understand how the cortical tracking of speech envelope is affected by stimulus modification, in particular inserting silent pauses. We hypothesise that the cortical responses would show increase in speech envelope tracking when pauses are added to the stimulus, especially the responses following onsets. This was formulated based on the studies by Hamilton, Edwards [1] and Chalas, Daube [26], where the authors reported that cortical response within 200 ms following silent pauses in the speech stream are relatively stronger than later sustained responses. In addition, the continuous speech with additional pauses is becoming more similar to the stimulation of repeating short sound (sound-silence-sound-silence) in ALR measurements, which onset responses may be generated more consistently compared to natural speech. The hypothesis appears to contradict with the previous findings reported by Kayser, Ince [22] and Hambrook, Soni [23], as the two studies showed that irregular speech rhythm (due to change in existed pause durations) and interruption of speech by silent pauses can reduce the listener's cortical tracking of speech. However, the stimulus manipulation method in the current study is different from the previous studies, the amount of silent pauses in speech increased considerably and speech sounds were not replaced by silent pauses.

One of the best established tools for measuring human neural response to speech stimuli is the Multivariate Temporal Response Function (mTRF) [6], which is used to quantify the linear relationship between sensory stimulus and its corresponding neural response. Two approaches can be employed with the mTRF (Fig 1), either encoding (predicting the EEG using the speech envelope) or decoding (reconstructing the speech envelope from the EEG) to estimate the cortical tracking of speech envelope. While the former follows the causal psychophysiological process of speech driving cortical tracking, the latter has practical advantages in allowing multiple EEG channels to be analysed simultaneously and thus potentially permits more powerful analysis of the association between speech envelope and a set of EEG signals than repeated single channel analyses. Considering these advantages, the decoding approach, will be implemented for data analysis throughout this study. The increase in envelope reconstruction accuracy,

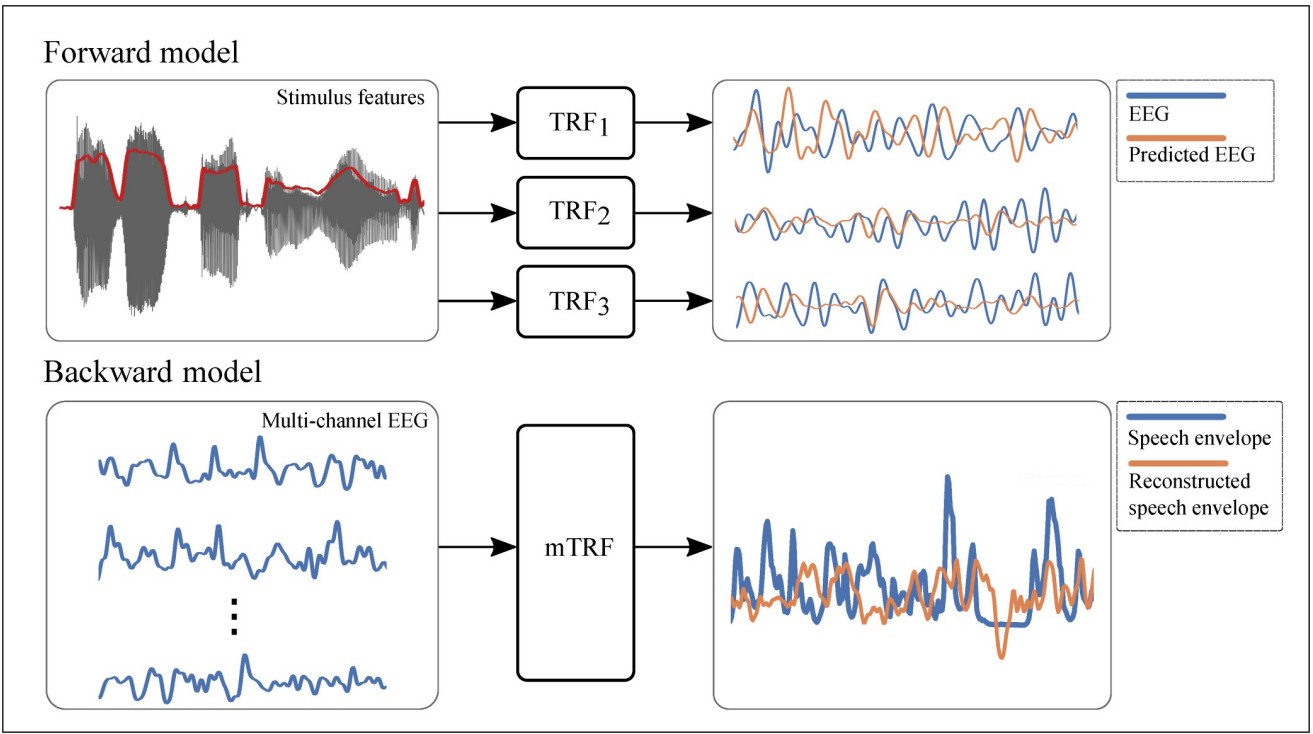

**Fig 1. Diagram of the temporal response function estimation in the forward and backward modelling approach.** The forward modelling approach or encoding model (TRF) uses a stimulus feature (i.e., speech envelope plotted in red) to predict the EEG response to a stimulus (here three TRFs and their associated EEG channels are shown). The backward modelling approach or decoding model (mTRF) uses EEG response to reconstruct the stimulus feature.

quantified by the correlation between the actual and the reconstructed envelope, indicated that the acoustic representation and the neural activity are more synchronous, thus more information may be parsed in the brain for processing of speech sound [7]. Details on the calculation of the backward mTRF (hence will be referred to as the decoder) will be explained in the methods section.

EEG was analysed in the delta and the theta band to explore whether the responses in these two frequency bands would reflect the difference in auditory processing time scales and roles, which is frequently reported in other studies [2, 22, 27]. For example, the cortical tracking of speech envelope in the delta band reflects the processing of words and phrases and tends to be strongly correlate with the intelligibility of speech [4, 27], while the cortical tracking in the theta band reflects the processing of syllables and is more correlated with acoustic features influencing speech segmentation [22, 27]. We also analysed the decoder for different segments of the speech envelope, in particular comparing the decoder calculated from the entire speech envelope to that of just onset and non-onset regions. Finally, we examine whether the decoding performance is influenced by the cortical response to speech or simply due to more input data when pauses are added to speech, this is done by limiting the data for analysis to be the same in duration.

The results are expected to provide new insights for comparing the decoder from different speech corpora which may have different speeds of delivery and a range of silent pauses between words, and the extent to which the observed cortical responses can be deemed to be dominated by the onset.

## Materials and methods

This section will describe the original speech stimuli and the modified versions and will describe the acquisition of the EEG data from participants. The pre-processing of EEG data, the main analysis tool used (the decoder), and an outline the statistical analyses performed are also presented.

### Participants

Sixteen native English speakers participated in this study (9 males; aged 18–41 years, mean 25 years old; 13 right-handed). All subjects self-reported as normal hearing. Hearing thresholds were tested with pure-tone audiometry in a sound-proof room, using air conduction. Thresholds for all participants were below 25 dB HL in the frequency range from 250 Hz to 8 kHz. Ethics were approved by the University of Southampton Ethics Committee (ethics reference number is 20741). All participants provided written informed consent prior to the experiments.

### Stimulus

The continuous speech stimulus used in this study was a segment of an auditory recording narrated by a female narrator in the free audiobook "The Children of Odin: Chapter 2—The building of the wall", available online at https://librivox.org/the-children-of-odin-by-padraic-colum/. The speech stimulus was manually split into four segments. Segments varied slightly in length to fit in with natural breaks, but each was approximately 2 minutes and 30 seconds in duration. In addition to using the recorded speech directly, the recording was also modified by inserting either 250 ms (short), or 500 ms (long) pauses between words, resulting in three speech pause conditions (natural speech, short, and long pauses). Speech unit were qualified as words based on written spelling (orthographic forms) [28], contractions were considered as one word. The length of each segment with short and long pauses inserted was approximately 4 minutes and 20 seconds, and 6 minutes respectively. A total of 12 segments of speech were used to test each participant, four segments per speech condition. The total duration for continuous speech stimulus for the natural speech, short pause, and long pause conditions were therefore approximately 10 minutes and 20 seconds, 16 minutes, and 21 minutes and 42 seconds, respectively.

Fig 2 shows the modulation spectrum of stimuli used across the three speech pause conditions. The peak modulation frequency of the stimuli occurs within the range of approximately 4–5 Hz. The modulation spectrum of stimuli used in the current study appears to be similar to the results reported by Ding, Patel [29]. Note that the modulation spectrum of speech with short and long pauses showed an additional peak at lower frequencies (delta band, 1–4 Hz), whereas the modulation spectrum of natural speech does not.

### Experimental procedures

The experiment was carried out in a quiet sound-proof room with lights turned off. Participants sat on a comfortable chair in a relaxed position and were instructed to close their eyes during the experiment to reduce ocular movement and consequent artefacts. They could take a break during the test if needed.

Participants were presented with the 12 segments of speech containing natural speech, speech with short, and long pauses (henceforth referred to as speech pause conditions). The speech pause conditions were presented in a randomised order to reduce order effects, but the 4 blocks within each condition were presented in a chronological order to maintain the flow

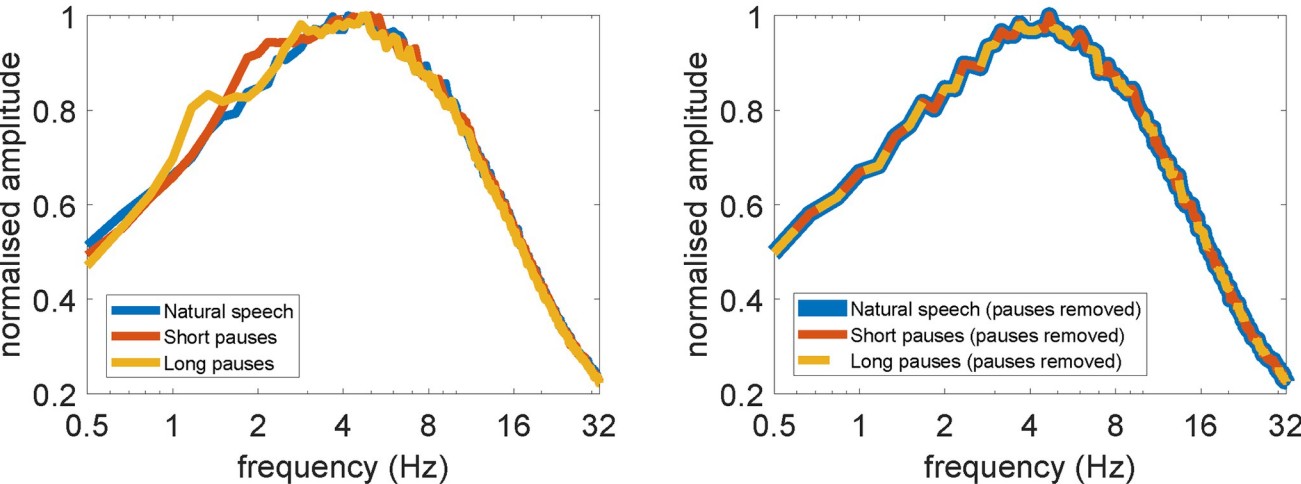

**Fig 2.** (Left) The modulation spectrum of the natural speech, speech with short pauses, and speech with long pauses stimuli. (Right) The modulation spectrum of the three stimuli completely overlaps after removing both natural and inserted pauses.

and progression of the story. Each participant was asked a multiple-choice question at the end of each speech segment, to assess their attention to the stimuli. However, the results from the behavioural task is unsuitable to be related to cortical responses pattern, as the questions were the same across the speech pause conditions. All 12 segments of speech were presented at 70 dB LeQ(A) (calibrated via Type 4230, Bruel and Kjaer), through ER-2 insert phones (Etymotic, Elk Grove Village, IL) to both ears.

EEG data were recorded using a 32-channel BioSemi EEG system (ActiveTwo, BioSemi BV, Amsterdam, Netherlands). Electrodes were positioned according to the international 10–20 system. Additional external electrodes were placed bilaterally on the mastoid and on the chin as reference channels and to detect artefacts from swallowing. The sampling rate for the EEG data was 4,092 Hz. A notch filter at 50 Hz was applied during data collection.

Analysis of EEG and speech stimulus were performed with the MNE-Python software [30]. EEG data from every participant were re-referenced to common average. They were then band-pass filtered using a zero-phase (non-causal) FIR filter (filter length was 6.6 times the reciprocal of the shortest transition band) over the range 1–4 Hz (delta band) and 4–8 Hz (theta band), and then resampled to 128 Hz. The EEG recordings from each participant were normalised prior to decoder analysis, to give a mean of zero and a standard deviation of 1.

### Extraction of speech envelope

The speech envelope was extracted using the Hilbert transform applied to the original speech stimuli (sampling frequency 44.1 kHz). The envelope was then band-pass filtered using the same filter settings as in the EEG analysis in the ranges 1–4 Hz and 4–8 Hz (matching the delta and theta frequency bands used in the EEG also—see below) and resampled at 128 Hz.

Prior to the analysis of the acoustic onset effect on cortical tracking of speech envelope via the decoder, in addition to the standard speech envelope used for model training, we processed onset and non-onset segments separately. To construct these segments, we detect the pauses in the speech envelope by setting a low threshold level around the zero value, then replaced the samples below the threshold level with Not a Number (NaN) to exclude them from further analysis. Pauses are excluded from both the onsets and non-onsets representation prior to the

decoder analysis, to avoid confounding from having different lengths of segments without any sound stimulus. In order to selectively process onsets, only samples within the first 150ms following each pause were kept, other samples (the non-onsets) were replaced with NaNs. This 150ms window was selected based on the duration of syllables as suggested by other studies [31–33]. To select only the non-onsets, the process was reversed, the onset segments samples in the speech envelope were replaced with NaNs and the remaining samples left with their original value. The full envelope refers to the data including onsets, non-onsets and pauses. Henceforth, the full envelope, onsets, and non-onsets will be referred to as the speech features. These extracted speech features are acoustic signal assumed to be encoded in the EEG, in which the two signals will then be analysed through the decoder.

## The multivariate temporal responses function (mTRF)

The backward mTRF, also known as the stimulus reconstruction or decoding approach, is a model-based approach which utilise a linear model ($g(\tau,n)$ in Eq (1)), to reconstruct the speech feature from the EEG response signal [34], using multichannel convolution:

$$\hat{S}(t) = \sum_n \sum_\tau r(t + \tau, n)g(\tau, n) \tag{1}$$

where $\hat{S}(t)$ refers to the reconstructed speech feature (in the delta or theta band) and $r(t + \tau,n)$ to the EEG signal (similarly filtered), and $t$ is the time index, $\tau$ is the range of time lags in the convolution (corresponding to model order), and $n$ represents the channel of the EEG. Eq (1) represents the so-called 'inverse model', since in reality the speech signal causes changes in the EEG, but in this model, the speech feature is estimated from the EEG using a non-causal filter (hence the + sign in $r(t + \tau,n)$). The linear model, referred to as the decoder, is calculated using the regularized least squares method,

$$g = (\boldsymbol{R}^T\boldsymbol{R} + \lambda\boldsymbol{M})^{-1}\boldsymbol{R}^T S \tag{2}$$

where $\boldsymbol{R}$ is the EEG signal in lagged time series (in matrix form), and $S$ is the stimulus feature (vector). The regularization matrix $\boldsymbol{M}$ is chosen to reduce the off-sample error (i.e. the error in predicting $\hat{S}(t)$ on previously unseen data) [35], as follows (missing terms are zero):

$$\boldsymbol{M} = \begin{bmatrix} 1 & -1 & & & & \\ -1 & 2 & -1 & & & \\ & -1 & 2 & -1 & & \\ & & \ddots & \ddots & \ddots & \\ & & & -1 & 2 & -1 \\ & & & & -1 & 1 \end{bmatrix} \tag{3}$$

Following the approach used in previous works [34, 36], Pearson's correlation was used to assess the ability of the decoder to reconstruct the measured speech feature. The decoders were validated using the leave-one-out cross-validation method. For each speech feature (full envelope, onsets, and non-onsets), a decoder was trained on three out of four speech segments used in each speech pause condition (natural speech, short or long pauses) and tested on the remaining segment. This process was repeated until the decoder was trained on all possible combination of three speech segments and tested on all speech segments within a speech pause condition. The resulting four Pearson's correlation coefficients and mean squared error (MSE) between the actual and the reconstructed speech envelope were then averaged. For each signal

type, the decoding process was repeated using different $\lambda$ parameters which were chosen from the range of 50 logarithmically spaced points between 0.01 to $10^{12}$. The optimal $\lambda$ parameter that gives the highest correlation coefficient after averaging the four Pearson's correlation coefficients was selected for each participant's decoder.

To test our hypothesis that the cortical response to speech with pauses would be enhanced by the onsets following pauses, we trained decoders on the full envelope as well as the onset and non-onset segments; thus, three decoders were produced for each speech feature (full envelope, onsets, and non-onsets) and each participant. For the samples where the speech envelope was set to NaN, the error in model fit cannot be calculated and they are thus excluded from the least-mean-square fitting in Eq (1). Then we tested each decoder on all three speech features to assess its ability to generalise across different parts of the recording. If the onsets dominate the EEG response and hence the decoder, then the decoder derived from the onsets should be able to predict the full envelope better (i.e., give a higher correlation coefficient) than the decoder derived from the non-onsets. Similarly, in this case one might also expect that the decoder derived from the full envelope would predict the onset responses better than that for the non-onsets. By testing each of the three decoders on all speech features, nine correlation coefficients are obtained, which can provide insight into which aspect of speech may be dominating the EEG response.

The recordings with pauses have a longer duration, as described in the "Stimulus" section. Comparison of decoder envelope reconstruction performance may be biased by this difference in length of recording. We therefore carried out an additional analysis to compare the Pearson's correlation coefficient across the different stimulation conditions using only 2 minute and 1 minute segments from each stimulus, with the same length of recording used in cross-validation.

## Statistical analysis

Permutation tests were performed to assess the significance of the correlation coefficients obtained from the decoders. A null distribution of correlation coefficients for individuals in each speech condition was obtained by using the speech envelope and a mismatched (permutated) EEG segment to train the decoder and perform the cross-validation on (previously unseen) testing data with the speech envelope and EEG segment also mismatched. Randomisation was repeated 500 times to construct the null distribution of Pearson's correlation coefficients. The correlation coefficient from the correct matching of speech envelope and EEG segment was then tested against this null distribution, at a significance level of $\alpha = 0.05$. In this way the significance of estimated correlation coefficients was tested in each recording, and not just of the average performance across the cohort.

Friedman tests were used to explore differences in the correlation coefficient within the same decoder training and testing combination across three speech pause conditions (multiple tests on natural speech, short, and long pauses conditions). Wilcoxon signed rank tests were used to explore differences in the correlation coefficients between each decoder training and testing combination. Bonferroni corrections were applied in all multiple comparisons. Adjusted $\alpha$-level after Bonferroni correction is 0.0167 (0.05/3) across three speech pause conditions for comparison within the same decoder training and testing combination. Adjusted $\alpha$-level after Bonferroni correction for tests within each speech condition is 0.00185 (0.05/27) for pairwise comparison, resulting from the nine different combinations of training and test datasets used (27 Wilcoxon tests in total for each EEG frequency band). Results were reported as statistically significant only in accordance with this Bonferroni correction, when $p \leq \alpha/N$.

## Results

Fig 3 shows the correlation coefficient between the true speech envelope and that reconstructed via the decoder, obtained from different decoder training and testing combinations, as a function of the pauses used (natural speech, short, and long pauses) in the delta and theta bands. Although nine correlation values were obtained for each speech pause condition (three training conditions and three testing conditions), for the sake of clarity only the results for training on the full envelope are shown, with others provided in the supplementary material (S1 and S2 Figs). We displayed 3 combinations of training and testing data on the decoder in Figs 3 and 4: 1.) decoder trained and tested on the full envelope, 2.) decoder trained on the full envelope and tested on onset regions only, and 3.) decoder trained on the full envelope and tested non-onset regions only. The increasing duration of pauses generally raised the correlation coefficient for all decoding features in both delta and theta bands. Friedman tests indicated statistically significant difference in correlation coefficients for comparison within the same decoder training and testing combination across the three pauses conditions ($p<0.001$ for the Bonferroni corrected level for significance $p \leq 0.0167$). The results of pairwise Wilcoxon signed rank test between pairs of speech features across all speech pause conditions are shown in Table 1.

In order to analyse the results in more detail, we will now consider first the decoder trained and tested on the full envelope, and then the decoder trained on the full envelope tested on the onsets, and finally the decoder trained full envelope tested on the non-onsets.

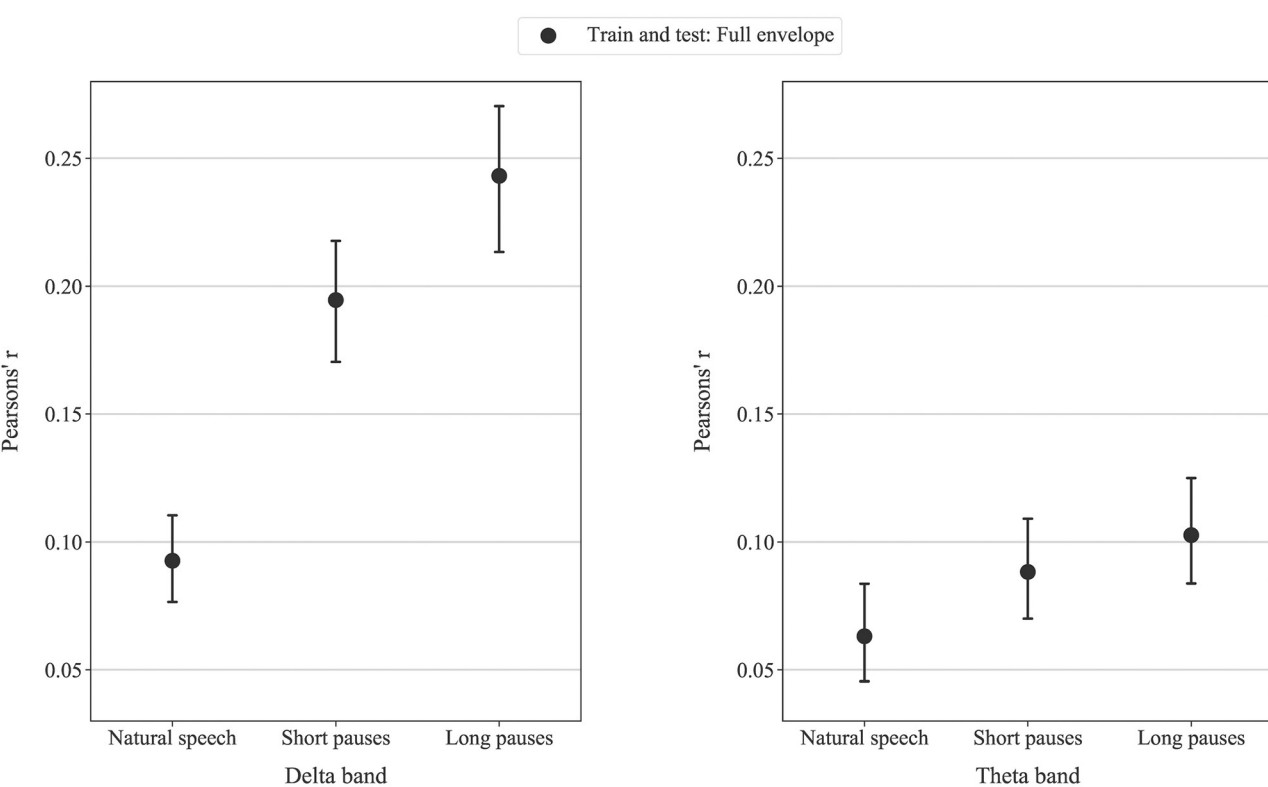

**Fig 3. The mean correlation coefficient from decoders trained and tested on the full envelope in (left) delta and (right) theta bands across three speech pause conditions.** Error bars indicate the 95% confidence interval for the mean.

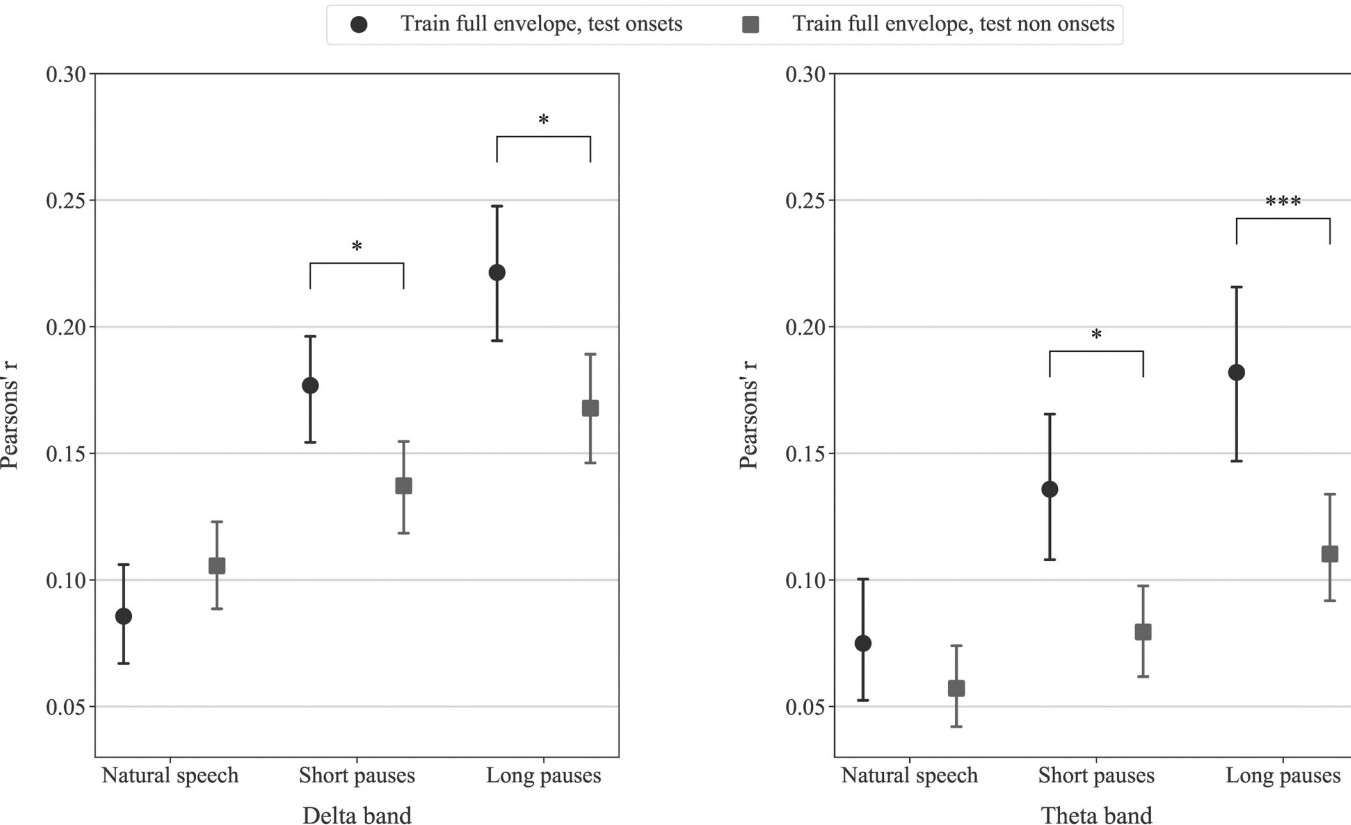

**Fig 4. The correlation coefficient from decoders trained on the full envelope and tested on onsets (circles) and non-onsets (squares) in (left) delta and (right) theta bands across three speech pause conditions.** Each point shows the average Pearson's correlation coefficient across sixteen participants. Error bars indicate the 95% confidence interval for the mean. Asterisks above paired points indicate significant differences in correlation coefficients (* for p<0.01 and *** for p<0.001).

## Effects of extended duration of pauses in continuous speech on the decoder trained and tested on the full envelope

From Fig 3, we observe that for the decoders trained and tested on the full speech envelope, the correlation coefficients gradually increase across the speech pause conditions in both the

**Table 1. P-values for all possible pairwise tests (Wilcoxon signed rank tests) across all speech pause conditions and speech features tested using model trained on the full envelope for both the delta and theta bands.** Significant p-values are shown in bold and italic (critical values from Bonferroni correction). P-values which are underlined indicate that the speech feature labelled at the top of the column with an underline has significantly greater correlation coefficients, or else the other speech feature is greater.

| Speech pause condition | Speech feature comparison pair | | |
|---|---|---|---|
| Delta band | F̲ull/Onsets | F̲ull/Non-onsets | Onsets/Non-onsets |
| *Natural speech* | 0.26 | **<*0.001*** | 0.02 |
| *Short pauses* | 0.013 | **<*0.0001*** | **<*0.001*** |
| *Long pauses* | 0.006 | **<*0.0001*** | **<*0.001*** |
| Theta band | F̲ull/Onsets | F̲ull/Non-onsets | Onsets/Non-onsets |
| *Natural speech* | 0.004 | 0.039 | 0.004 |
| *Short pauses* | **<*0.0001*** | 0.07 | **<*0.001*** |
| *Long pauses* | **<*0.0001*** | 0.011 | **<*0.0001*** |

delta and theta bands (p<0.001, Friedman tests). The improved reconstruction of the full speech envelope indicates that neural responses to speech with extended pauses are better aligned with the speech envelope (i.e., enhanced linear relationship to the speech envelope), as originally hypothesised.

A similar trend of increasing correlation coefficients for the full envelope decoders is also shown in the theta band (Fig 3 right) (p<0.001, Friedman). This further reinforces the strong influence of additional pauses in speech to the cortical envelope tracking. It may also be noted that the correlation coefficient from the delta band is higher than for that the theta band. This agrees with previous studies using the stimulus reconstruction approach [27, 37].

### Effects of extended duration of pauses in continuous speech on the decoder trained on the full envelope tested on the onsets and non-onsets

From Fig 4, when using the decoder trained on the full envelope, the correlation coefficients for testing on onsets or non-onsets were not significantly different in the natural speech condition. However, when short or long pauses were included in the speech, the correlation coefficients of decoder tested on onsets were significantly greater (p<0.001) than for decoder tested on non-onsets, indicating that the decoder is better adjusted to the onset than the non-onset speech envelope segments. This suggests that with the longer pauses, the cortical tracking of speech envelope becomes dominated by the onsets. Similar impacts of pauses in the speech on the decoder tested on onset and non-onset segments were observed in both delta and theta bands, though more dramatically so in the delta band (Fig 4 left). The higher correlation coefficients achieved when testing the decoder on the onset segments compared to the non-onset segments are very clear, with statistical significance, in accordance with our original hypothesis.

### Comparing significance of cortical tracking of speech with extended pauses using different amount of testing and training data

Fig 5 shows the Box and Whiskers plot of decoder correlation coefficients across subjects using EEG data with different durations per segment across the three speech pause conditions in both the delta and theta band. With the same amount of training and testing data, the cortical tracking of speech with additional pauses inserted between words generally show significantly stronger correlation (see Table 2) compared to cortical tracking of natural speech (p<0.001), except when comparing correlation coefficients between response to natural speech and short pauses condition in the theta band. This implies that the increase in correlation coefficients in the short and long pauses conditions is not simply a result of having longer EEG recordings trained on the decoder.

### Onset and non-onset segments sample amplitude distribution

Fig 6 shows the distribution of sample amplitude of onset and non-onset segments in the delta and theta band across the four speech segments. In the current study, we only show the results from the decoder trained on the full envelope and tested on the full envelope, onset, and non-onset segments. We were initially concerned that onset and non-onset segments may have a different amplitude range and since larger amplitude ranges tend to lead to increased correlation coefficients, such differences might bias results. However, further analysis showed that onset and non-onset segments had similar speech envelope amplitude ranges, reducing this concern. It may also be noted that the non-onset segments contain greater number of samples

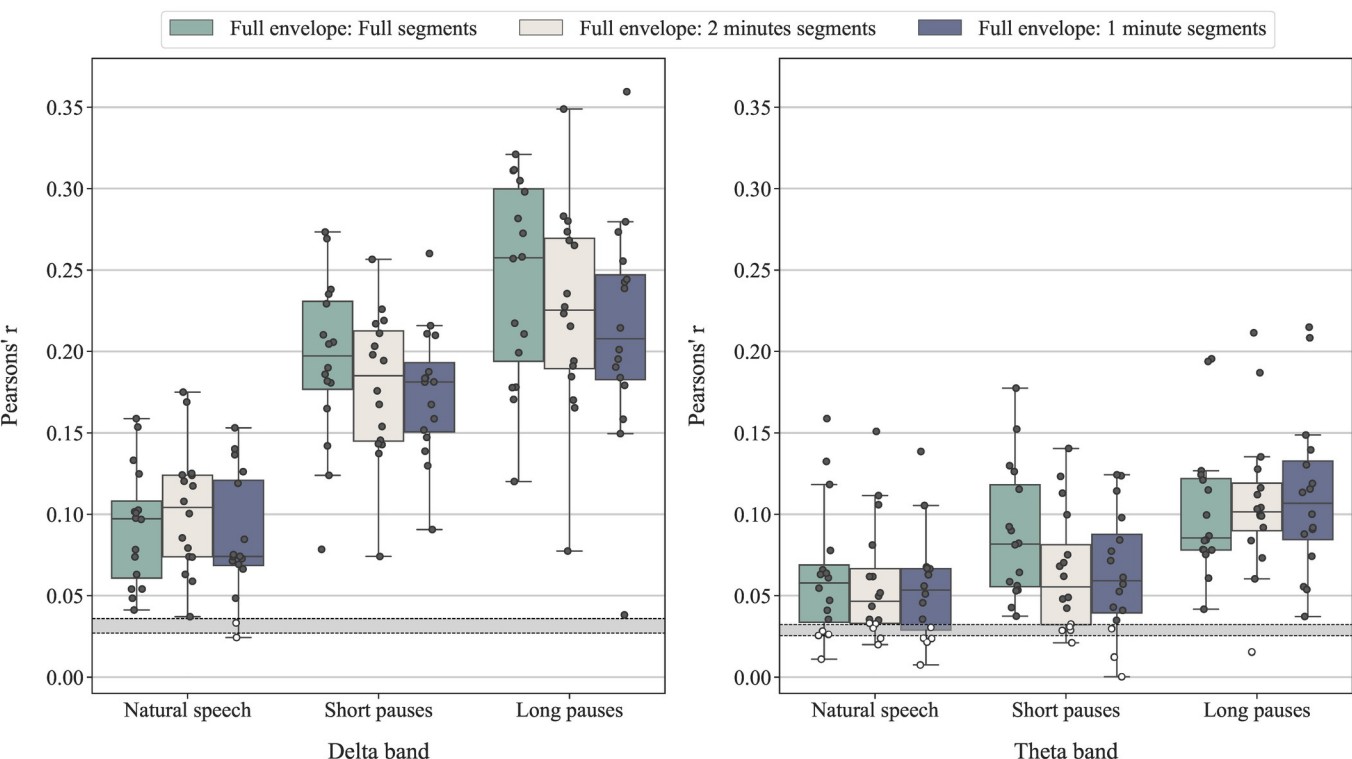

**Fig 5. Box and Whiskers plot of correlation coefficients from each participant's decoder using different amount of training and testing data in the delta (left) and theta band (right).** Light grey, grey, and dark grey boxes contain correlation coefficients from the full envelope using different stimulation durations (full length, 2 minutes, and 1 minute segments) recording from each data segment (in total 4 segments), respectively. Full segment stimulation refers to the full duration of the recording of each segment including natural pauses, whereas the 2 and 1 minute segments stimulation refers to recording segments with added pauses whose duration is truncated to 2 or 1 minute, respectively. The grey horizontal band indicates the range of critical values obtained from individuals in the sample, based on the null distribution of the correlation coefficients only from decoders trained and tested on the full envelope with segments in full length (i.e., all estimates above this band are deemed significant). Dots overlaid on each box are the decoder correlation coefficients from each participant. White dots indicate individual correlation coefficients that are not statistically significant based on subject's null distribution.

than the onset segments. This implies that the increased in correlation coefficients from the decoder was neither a result from greater amplitude variance in the samples of different speech feature segments nor bias towards a model which was trained on greater amount of data because all the models were trained on the full speech envelope.

**Table 2. P-values of differences in correlation coefficients between different stimulation durations (Wilcoxon signed rank test).** Bold and italic p-values indicate statistically significant difference (Bonferroni corrected) in correlation coefficients between data reduction conditions.

| Segment lengths | Natural speech and short pauses | Natural speech and long pauses | Short pauses and long pauses |
|---|---|---|---|
| Full (delta) | *<0.0001* | *<0.0001* | *<0.001* |
| 2 minutes (delta) | *<0.0001* | *<0.0001* | *<0.002* |
| 1 minute (delta) | *<0.0001* | *<0.0001* | 0.049 |
| Full (theta) | 0.034 | *<0.001* | 0.015 |
| 2 minutes (theta) | 0.679 | *<0.001* | *<0.001* |
| 1 minute (theta) | 0.179 | *<0.0001* | *<0.001* |

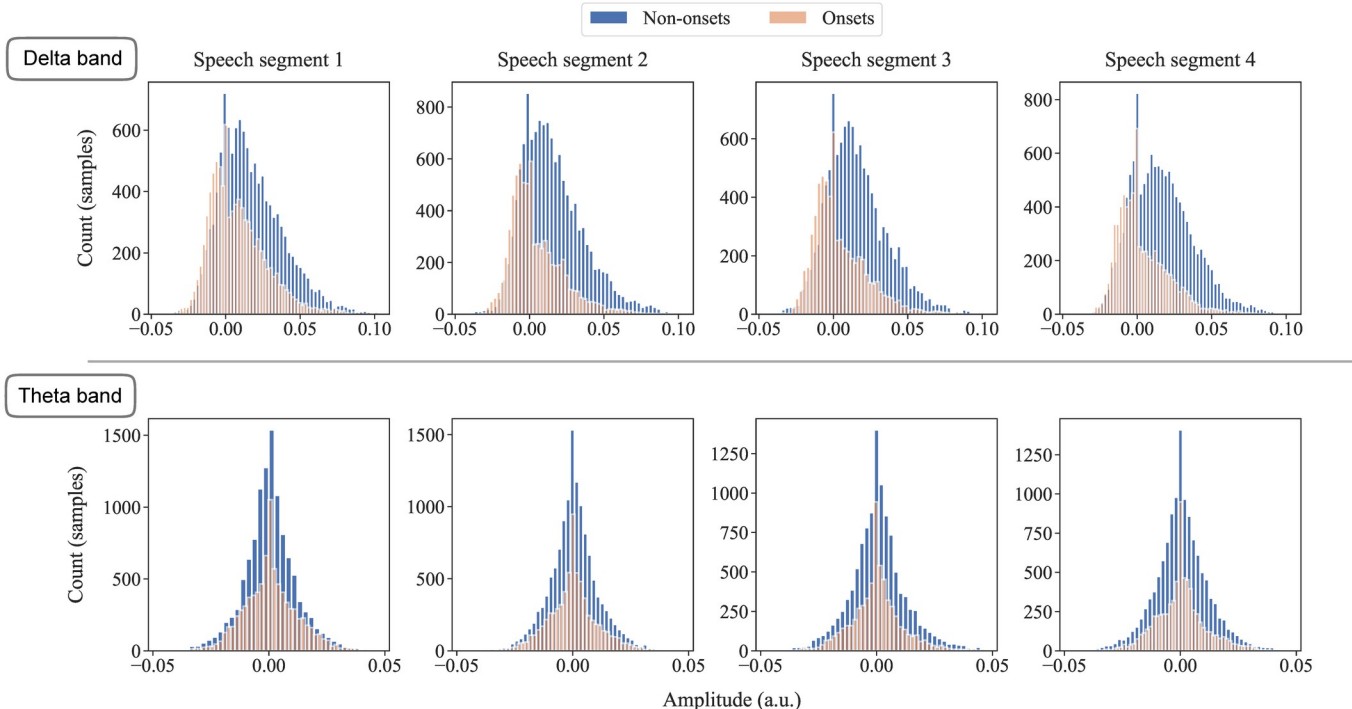

**Fig 6. Amplitude histogram of onset (orange) and non-onset segments (blue) in the delta (top row) and theta frequency bands (bottom row) across the four speech segments.**

### Effect of delta band acoustic modulation on the cortical tracking of speech envelope

As shown in Fig 2 that an additional peak in the modulation spectrum appears in the delta band frequency for speech with short and long pause inserted, it is unclear whether the relatively greater correlation coefficients in the delta band compared to the theta band was a result of the delta modulation rate or not. An additional analysis was conducted by removing samples in pause segments and samples in the EEG in lagged time series at the same sample index from the decoding process. This was done to only relate the EEG to the envelope where speech occurs, ensuring a consistent modulation spectrum across different speech pause conditions. This will be referred to as the pauses removed condition. We also included an additional pause removed decoding condition using longer EEG time lag, 0–500 ms, to examine if the original 0–300 ms time lag was sufficient to capture the effect of the delta band modulation rate.

Fig 7 shows the mean correlation coefficients averaged across all participants obtained from decoders trained and test on the full envelope and the envelope with pauses removed in three pauses conditions. Specifically for this analysis, the adjusted significance level for multiple comparisons was adjusted to $p \leq 0.0056$ for 9 pairwise comparisons (within each speech pause condition only) in each EEG frequency band. Overall, the same trend in increasing correlation coefficients when longer pauses were inserted to the speech stimulus as reported earlier remains but the correlation values changed significantly when pauses were removed from the decoding process. In the delta band, in the short and long pauses condition, the correlation coefficients when decoding with pauses removed were significantly lower compared to when decoding using the full envelope ($p < 0.001$). While in the theta band, correlation coefficients from

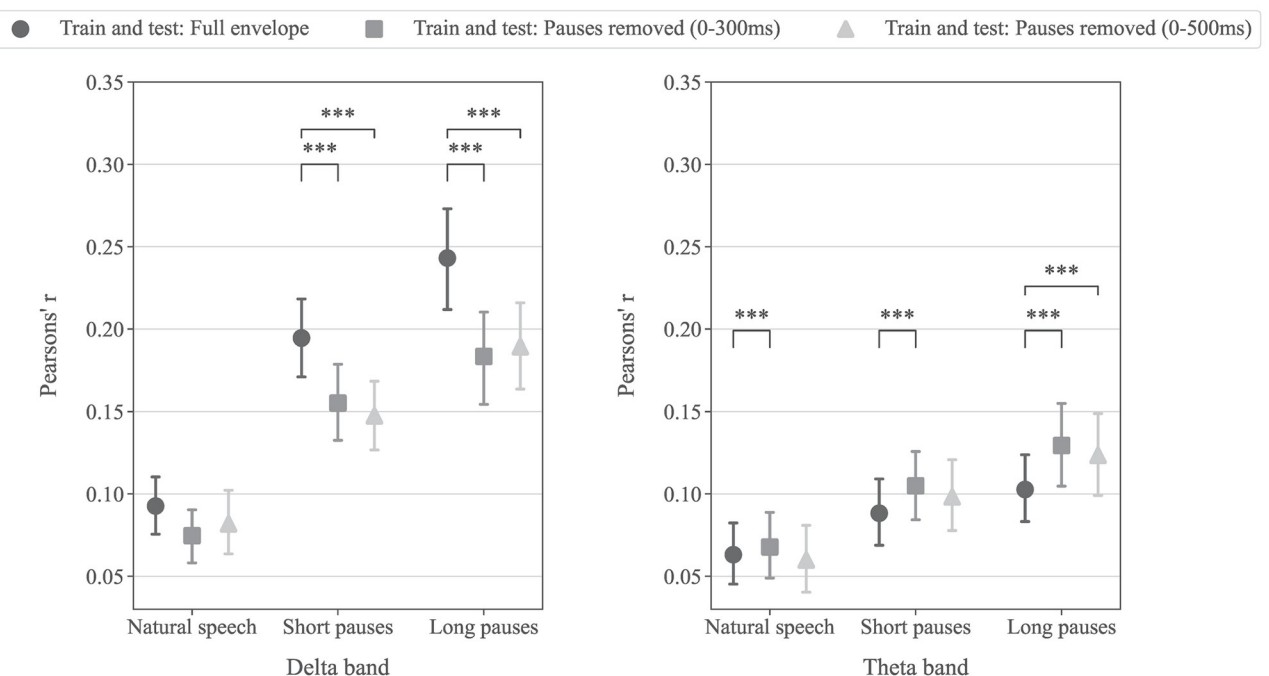

**Fig 7. The mean correlation coefficient from decoders trained and tested on the full envelope and envelope with pauses removed in (left) delta and (right) theta bands across three speech pause conditions.** Each point shows the average Pearson's correlation coefficient across sixteen participants. Error bars indicate the 95% confidence interval for the mean. Asterisks above paired points indicate significant differences in correlation coefficients (*** for p<0.001).

decoders with pauses removed were significantly greater than when decoding using the full envelope (p<0.001). Correlation coefficients from pauses removed decoders with different time lags, 0–300 ms and 0–500 ms, were not statistically significant in both the delta and theta band.

The greater correlation coefficients from the decoder in the delta band compared to the theta band appear to be partially affected by the delta band modulation rate in speech with inserted pauses. This is due to the significant decrease in correlation coefficients in the delta band when the pause segments are removed from the decoder analysis, as the EEG delta oscillation may persist in those segments. Despite this, the trend of increasing cortical tracking when pauses are inserted in speech remains. It is also evident that the effect of inserted pauses in speech is stronger for the cortical tracking of speech envelope in the delta band than in the theta band.

## Discussion

This study has demonstrated that cortical auditory evoked responses to continuous speech with additional pauses inserted between words show increase in tracking of speech envelope relative to responses elicited by natural speech. These results are based on decoder analysis using Pearson's correlation between the reconstructed and the actual speech envelope to test model fit. It also appears that the onset response to speech contributes more to the improved speech envelope reconstruction when pauses are introduced into the stimulus. This finding is consistent with previous studies suggesting that the auditory cortex is sensitive to acoustic edges [3, 38, 39].

Modifications in the speed of presented speech have typically been carried out in previous studies by either modifying silent pauses alone, or by altering (compressing or expanding) the

temporal waveform of speech. An advantage of only modifying the duration of pauses, as used in this study, is that its effect on speech intelligibility can be investigated independently from effects of changes in acoustical properties, such as intensity, frequency, and the speech envelope of individual words. The main disadvantage of manipulating pauses in speech is that the flow of speech can be severely altered when inserting pauses between words or phrases and in the current case the speech does indeed sound quite unnatural with the inserted pauses. The advantage of using time-compressed or -expanded speech is that the overall rhythm of speech does not change greatly compared to natural speech, however there are changes in acoustic properties affecting the articulation of phonemes [20]. Analysis of EEG responses to compressed or expanded speech may lead to confounding between the effects of changes in pauses and changes in phonemes. Our experimental protocol only affected the pauses and clearly demonstrated their powerful effect on EEG responses.

The longer duration of the stimuli in the short and long pauses conditions may increase the decoding performance due to more training data, however, we have demonstrated in Fig 5 that the increase in stimulus envelope reconstruction accuracy is mainly due to the pause effect. In Fig 5, when the decoder was trained and tested using an equal amount of data across speech pause conditions, the consistent trend of increasing correlation coefficients with longer pauses in speech persists.

## Effect of pauses in speech to cortical auditory responses

As shown in Fig 4, the pauses in speech not only increased the average correlation coefficients in the group but could achieve this with shorter recordings. The correlation was also statistically significant in each subject when using short and long pauses, but only in 9 out of 16 subjects when using natural speech and training data of 3 minutes (segments of 1 minute in the leave-one-out cross validation) or 6 minutes (segments of 2 minutes). The use of shorter segment length not only lower the correlation coefficient value but also caused the critical values of the correlation coefficient obtained from the permutation test to be greater compared to the use of full recording, which leads to greater number of non-significant correlation coefficients.

Considering that the duration of the speech stimulus with additional long pauses expanded more than two times relative to the natural speech stimulus (from approximately 10 minutes to 21 minutes), the intelligibility of the stimulus can change significantly. Although the additional pauses inserted to the stimuli did not alter the speech directly, as there was no time-compression or expansion applied on the temporal waveform, the added pauses may break the phrases or sentences structure and boundary, thus it becomes more difficult to comprehend the ongoing story. Previous studies have shown that the cortical tracking of speech envelope is positively correlated to the behavioural speech-in-noise performance for normal hearing people [4, 37]. Due to the lack of behavioural data in the current study, a clear conclusion on how the increase in cortical tracking when pauses were added to speech relates to individual's speech comprehension cannot be drawn. However, it is important to consider that the cortical tracking of speech envelope alone may not be an ideal measure to indicate how well a person can understand speech, as it can be influenced by both encoding of acoustic and cognitive processing related to speech comprehension, such as attention to target speech and effort in listening [24, 40]. For example, a listener might show greater cortical tracking of speech envelope when listening to speech in a language they cannot understand than when listening to a language they can understand [40]. Therefore, in this current study, the increase in cortical tracking when pauses were added to speech does not necessarily imply that the participants have improved speech understanding. Future studies may consider implementing the encoding

models to dissociate the contribution of lower-level (e.g., acoustic envelope) and higher-level (e.g., phonemes and phonetic features) information of speech to the cortical responses to speech with pauses [5].

There have been two previous studies that have explored the effects of stimulus manipulation on neural responses to speech, though the protocol was different to that used in the current study and the pattern of responses found was also somewhat different. Kayser, Ince [22] made a comparable study, investigating the effect of irregular speech rate on the neural and behavioural responses to speech. They found a reduction in the cortical tracking compared to natural speech only in the delta band, with no difference in other frequency bands. Behavioural speech intelligibility also remained approximately the same for both natural and irregular speech. It was suggested that the top-down processes of speech perception, using prior knowledge in language to comprehend speech [41], reduced cortical tracking in the delta band. Top-down processing has been found to be sensitive to the regularity of sound [42, 43] which might affect the cortical tracking. The reason that the cortical tracking of an irregular speech envelope in other frequency bands remain similar to that of cortical tracking of natural speech may be that the modification of pauses in the study by Kayser, Ince [22] was primarily controlled to preserve the overall mean duration of pauses. The modified duration of pauses only changed compared to their original duration (and was limited to a maximum of 300%), rather than consistently increasing, as was the case in our work. The duration of pauses in Kayser's work was probably not consistent or long enough to enhance auditory onset responses.

Another study that can be compared with ours is by Hambrook, Soni [23], who examined the effect of periodic introduction of pauses and noises into the continuous speech, on both behavioural responses and cortical tracking. The aim of their study was to explore neural function during the phonemic restoration phenomena, which refers to the observation that speech intelligibility degrades when the speech stream is interrupted by silent pauses and partly restored when noises fill the interrupting pauses. The introduction of pauses with the duration of 166 ms every 333 ms into the continuous speech (50% of speech removed) was found to significantly reduce speech intelligibility and the cortical tracking of speech envelope, however, speech intelligibility and cortical tracking of speech envelope improved when pauses were filled with noise. The disruption of acoustic tracking in the auditory cortex was suggested to be the reason for the reduction in cortical response. Although they suggested that it is possible that the onset and offset segments of the speech envelope can be removed and replaced by silence, causing disturbance in the cortical tracking on those segments, they found no evidence to support this. This reinforces the idea that speech comprehension is not a process which is driven by the stimulus alone. In their study, the actual speech was affected by interruption of pauses and noises, while in our study speech were not replaced by pauses and noises. We presume that their manipulation method might disrupt the cortical processing, as speech was removed, whereas our speech manipulation presumably did not and indeed increase the cortical tracking of speech envelope.

Future studies incorporating the mTRF paradigm should consider the potential onset effect when comparing measures of cortical tracking obtained from different speech stimuli. The cortical tracking to less natural speech containing more silent pauses, such as the Matrix sentences used for assessing speech reception threshold, may be more influenced by the acoustic onsets [37]. The objective measure values may then become unsuitable for direct comparison and interpretation, as they were estimated using responses and stimuli containing different acoustical properties. For example, the comparison objective measure might not clearly indicate whether one stimulus is more intelligible to the listener than the other. Another practical consideration when using the mTRF paradigm is that it may be suboptimal to apply a linear model trained on one type of speech and testing on a considerably different type of speech. For

example, training a model on narrative speech stimulus and testing on Matrix sentences stimulus, and vice versa.

### Differences in the methods to define onsets

The selection of onset segments in this study differed from previous studies using EEG response. In our work, the first 150 ms portion following word onset is defined as the onset segments. Previous studies commonly calculate onset envelopes from the first derivative of the speech envelope (gradient of the speech envelope e.g., [3, 36]. The gradient of the speech envelope only contains the rate of amplitude change, which is greatest for onset and offset segments. The reconstruction accuracy of the gradient of the speech envelope often results in weaker correlation compared to the reconstruction of the standard speech envelope [3, 36]. One problem with the gradient envelope is that it not only removes pauses, but also relatively constant amplitude sections of for example voiced speech. Our method of specific analysis of onsets using decoder does not appear to have been used previously in this area and seems better able to focus on these signal segments than previously used alternatives.

A study by Hamilton, Edwards [1] observed the effect of onsets in a similar manner. In their study, they found that ECoG responses following pauses longer than 200ms generates strong onset responses that can occur both within a sentence or before the sentence starts. Our study extended their findings, by demonstrating that effect of strong onsets response persists even when the pauses are 500 ms in duration and it could be detected by a non-invasive EEG measurement. It may be possible to investigate certain regions of the brain where they are specifically sensitive to acoustic edges and onsets non-invasively, but we have not specified whether the strong onset response from the EEG was generated from the same region (Superior temporal gyrus—STG), as shown in the ECoG studies.

Some studies may refer the cortical tracking of envelope to as the phase-locked responses to amplitude modulation, specifically phase-locking to change in acoustic cues [39, 44]. It was suggested that the strength of phase-locked responses to amplitude modulation may be associated with strong onset response [39]. Other study also found that the phase-locked responses are enhanced when stimulated with intelligible speech compared to less-intelligible speech with evidence of no onset response effect [44]. A confounding onset and intelligibility of continuous sound stimulus effect on the cortical responses has been presented through these studies. Thus, measurement of auditory responses to specific speech tokens or sounds, such as phonemes or consonants, may not be appropriate when aiming to probe auditory responses to higher level information in continuous speech [5].

A limitation of the study was the use of a fixed time interval (150 ms) to represent acoustic onset regions in speech. It should be noted that this time interval does not necessarily correspond with linguistic information. For example, the length of syllables varies and the unique time point at which words can be unambiguously identified will vary for different words. As a result, the study does have a potential confound between neural process representing acoustic onsets and those representing linguistic boundaries. An interesting area for future work could be to define linguistic boundaries in the speech stream and to compare tracking based on linguistic boundaries to those using fixed acoustic onset regions.

The different behaviours for onset and non-onset responses adds to the discussion on analysing AERs to speech: are we primarily observing the response to acoustic features or to higher level processing of speech? If the former, one might ask if speech is the most efficient stimulus to use and to what extent it provides additional information to that obtained by repeated transient synthetic stimuli or by speech tokens.

## Conclusion

Continuous speech with additional pauses inserted between words increases the cortical tracking of speech envelope in both the delta and theta band compared to using natural speech. Analysis of the way in which different components of the speech envelope are reconstructed from the EEG signal suggested that there are two distinct responses: onset and non-onset responses. Cortical responses to natural speech in the delta and theta band are not strongly related to either onset or non-onset segments, however, when pauses were introduced into the speech, responses in both frequency bands become dominated by the onsets.

The influence of the onset responses also led to increased number of cases where cortical tracking of speech envelope was significant, but this may not be an indication of better speech comprehension. The results clearly demonstrate how the correlation obtained from the decoder can be affected by acoustic characteristics of the selected speech, which has the strong potential to be a confounding factor in comparing studies and making inferences on speech intelligibility from decoding mTRF analysis.

## Supporting information

**S1 Fig. The correlation coefficient of each decoder training and testing combination in the delta band across three speech pause conditions.** Each point indicates the average Pearsons' r across sixteen participants. Error bars indicate the 95% confidence interval for the mean. (TIF)

**S2 Fig. The correlation coefficient of each decoder training and testing combination in the theta band across three speech pause conditions.** Each point indicates the average Pearsons' r across sixteen participants. Error bars indicate the 95% confidence interval for the mean. (TIF)

## Author Contributions

**Conceptualization:** Suwijak Deoisres, Yuhan Lu, Frederique J. Vanheusden, Steven L. Bell, David M. Simpson.

**Data curation:** Yuhan Lu, Frederique J. Vanheusden.

**Formal analysis:** Suwijak Deoisres, Yuhan Lu.

**Investigation:** Suwijak Deoisres.

**Methodology:** Yuhan Lu, Steven L. Bell, David M. Simpson.

**Software:** Suwijak Deoisres, Frederique J. Vanheusden.

**Supervision:** Steven L. Bell, David M. Simpson.

**Validation:** Suwijak Deoisres.

**Visualization:** Suwijak Deoisres.

**Writing – original draft:** Suwijak Deoisres.

**Writing – review & editing:** Suwijak Deoisres, Steven L. Bell, David M. Simpson.

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
