## [Decision Letter · Decision Letter 0]

5 Dec 2022

PONE-D-22-18218Continuous speech with pauses inserted between words enhances cortical envelope entrainmentPLOS ONE

Dear Dr. Deoisres,

Thank you for submitting your manuscript to PLOS ONE. After careful consideration, we feel that it has merit but does not fully meet PLOS ONE’s publication criteria as it currently stands. Therefore, we invite you to submit a revised version of the manuscript that addresses the points raised during the review process.

We look forward to receiving your revised manuscript.

Kind regards,

Michael Döllinger, Ph.D.

Academic Editor

PLOS ONE

https://journals.plos.org/plosone/s/fileid=ba62/PLOSOne_formatting_sample_title_authors_affiliations.pdf.

Reviewers' comments:

Reviewer's Responses to Questions

**Comments to the Author**

1. Is the manuscript technically sound, and do the data support the conclusions?

Reviewer #1: Yes

Reviewer #2: Yes

2. Has the statistical analysis been performed appropriately and rigorously? 

Reviewer #1: Yes

Reviewer #2: Yes

3. Have the authors made all data underlying the findings in their manuscript fully available?

Reviewer #1: No

Reviewer #2: Yes

4. Is the manuscript presented in an intelligible fashion and written in standard English?

Reviewer #1: Yes

Reviewer #2: Yes

5. Review Comments to the Author

Reviewer #1: The manuscript reports an EEG experiment investigating the role of added pauses on the cortical tracking of speech. Specifically, as the neural measure for speech tracking, the authors focus on the reconstruction of the speech envelope from the EEG signal (forward/decoding multivariate Temporal Response Function model). The results show that added pauses yield higher envelope reconstruction values compared to natural speech, a finding that seems to be primarily driven by word onsets.

- Terminology: entrainment vs tracking

Considering recent debates in the literature (see refs below), I would suggest the authors to use the term “(cortical/neural) tracking of speech” instead of “entrainment”.

Obleser, J., & Kayser, C. (2019). Neural Entrainment and Attentional Selection in the Listening Brain. Trends in Cognitive Sciences, 23(11), 913–926. https://doi.org/10.1016/j.tics.2019.08.004

Meyer, L., Sun, Y., & Martin, A. E. (2020). “Entraining” to speech, generating language? Language, Cognition and Neuroscience, 35(9), 1138–1148. https://doi.org/10.1080/23273798.2020.1827155

- Goal of the study:

At the beginning of the Introduction, it is mentioned that the goal is to provide the mTRF counterpart of ALRs, while later (lines 137 – 139) the goal shifts to how added pauses modulate the cortical tracking of speech (in terms of forward mTRF). From my understanding, the latter seems to be the main focus throughout the whole manuscript, while the former is barely addressed later on. Could the authors revise this point and provide an explicit and consistent description of their goal(s) in the Introduction? By doing so, the authors could also elaborate more on the theoretical motivations of their goal beyond just its empirical justification.

- Expected results:

The authors hypothesize stronger cortical responses with added pauses (lines 139-140). However, they do not provide any explanation for this expected result, which might seem even contradictory considering the previous findings reported above (Kayser’s and Hambrook’s studies). Could the authors provide a clear motivation for the mentioned expected results in the Introduction?

- Rationale of mTRF approach:

Although the authors do specify in the methods section that they will follow the decoding/backward mTRF approach, the terminology used in certain parts of the manuscript is not unambiguous in this respect. In particular, the authors often refer to higher envelope reconstruction correlations as “stronger/enhanced cortical responses”, which might wrongly suggest that encoding/forward mTRF models were used (e.g., referring to higher TRF weights). Moreover, considering that envelope reconstruction correlations could be reflecting phase rather than (only) amplitude alignment between both signals, using terms such as “stronger” or “enhanced” might be misleading. Based on these issues, I suggest the authors (i) specify in the Introduction how mTRF models will be implemented and explain their main rationale (e.g., what does higher envelope reconstruction mean in terms of neural responses?) and (ii) revise how the (interpretation of the) results is conveyed throughout the manuscript.

- Stimuli:

The method section reports that stimuli were modified by adding short/long pauses between words. Which linguistic (e.g., orthographic/phonological/phonetic?) criteria were used to qualify a speech unit as a word? For example, were contracted forms (e.g., “we’re” instead of “we are”) considered as one or two words? This seems quite relevant considering the use of natural speech, which probably contained many connected speech parts. Providing detailed information about stimuli preparation would be very helpful to better understand the manipulation as well as considering future studies. Also, would it be possible to have access to some of the stimuli?

Considering the temporal nature of the manipulation, it would be very informative to provide the power spectrum of the stimuli (e.g., Fast Fourier Transform of the speech envelope) as usually done in neural tracking studies. If there were power differences in the speech envelope across conditions, how would they relate to the reported neural tracking findings?

- Behavioural responses:

The results from the behavioural task (comprehension questions) are not included in the manuscript. Could the authors report them? Also, could the authors somehow relate the patterns from the behavioural responses and the envelope reconstruction correlations?

- Language comprehension/intelligibility:

Although the authors sometimes make some reference to intelligibility issues, it would be interesting to elaborate more on this when explaining the results of the current manuscript. How does the current results relate to the mentioned mixed findings in the literature regarding the relationship between neural tracking and intelligibility? Addressing this important issue would also increase the theoretical implications of the reported results.

- Onset vs non-onset comparison:

In the methods section “Extraction of speech envelope”, it is said that onsets were taken from the first 150 ms of each word. Because of this approach, relevant linguistic aspects could significantly differ across words, which could somehow impact the reported results. For example, uniqueness point (the specific time point at which that word can be unambiguously identified), informativeness (e.g., how surprising/entropic is that word at that time in terms of phonetic/lexical information?) or syllabic chunking (does that speech segment corresponds to a single syllable or covers partial syllable(s)?) might vary considerably across words. From my perspective, these potential confounds represent limitations for the onset vs non-onset comparison as well as the interpretation of such results.

In addition, in lines 402-408, the authors mention a few control analyses/measures for the onset vs non-onset contrast. Could the authors report the results of these control analyses/measures?

Reviewer #2: Deoisres et al. explored how inserted pauses between words in natural speech stories modulate neural encoding/decoding of speech signals using a mTRF framework. The results indicate that the encoding/decoding performance is increased by inserted pauses between words.

I found this study interesting. It is interesting to know how the encoding/decoding performance is changed after one inserts pauses between words or slows down normal speech. For hearing-impaired or elderly people, they can process speech better if the speech is slower, which is probably their encoding of word onsets becomes better.

I am positive about this manuscript and here are my comments:

1.The authors do not clearly explain why the study was done. It is difficult to understand the meaning of the study for readers who are not familiar with this field. The interpretation of the data results is not very comprehensive, only the data results are presented, but the interpretation is very limited. More details can be added so that uninitiated readers can find this manuscript more accessible?

2.It seems to me that: The authors seem to confuse, sometimes, between the mTRF technique and the continuous speech processing paradigm, and it is felt that the authors did not express the difference between the two clearly. Can the authors make this difference more clear?

3.Can the authors spell out ALRs when it first appeared in the manuscript?

4.Can the authors elaborate more on how they went from ALRs to mTRF? I understand this logic from my knowledge, but, again, the authors seem to assume that most of readers should understand this, which seems hard just by reading the manuscript.

5.The total duration for continuous speech stimulus for the no pause, short pause, and long pause conditions were therefore approximately 10 minutes and 20 seconds, 16 minutes, and 21 minutes and 42 seconds, respectively. Does the length difference here affect the encoding/decoding performance? The amount of data sometimes affects values of correlations. Can the authors provide some thoughts on this?

6.By the authors' own description in their article, the growth multiplicity of the story has exceeded 0.5 under long pause conditions, which significantly affects the intelligibility of the story, and why the response of mTRF is enhanced under such speech that affects intelligibility, the authors need to explain more on this point.

7.Line 139. “We hypothesize that the cortical responses to continuous speech would become stronger when pauses are added to the stimulus, especially the responses following onsets. “ I haven’t quite understood this hypothesis. Maybe it is my negligence, but if the authors can make this hypothesis easy to understand, that would help readers in my opinion.

8.I do not understand the application implications here, and perhaps the authors could provide more detail on the value of comparing the mTRF from different speech corpora.

Minor comments

1 The authors lacked a detailed explanation of what mTRF detection time is and the relevant references.

2 ‘mTRF was designed to measure the strength of envelope entrainment. ‘

The mTRF does not seem to be specifically designed to measure the strength of envelope entrainment, but has a broader use and more general principles, and if the authors insist on this opinion, they should add relevant references.

3 ‘As it is known that the EEG can entrain to the envelope of speech [7], the mTRF was designed to measure the strength of envelope entrainment. ‘

[7] This article is still based on research under the traditional ERP or isolated stimulus paradigm, and there are many references to mTRF paradigm research.

4 ‘Randomization was repeated 100 times to construct the null distribution of Pearson's correlation coefficients. ‘

The 100 times of permutation test is a bit low, and it is recommended to try more permutations (500 or 1000) to obtain more stable results.

5 The results of all paired tests might be better presented together with the figures, as two separate presentations may make it difficult for the reader to read and integrate all statistical results. Just a thought.

6 There are several typos and some sentences do not sound right. It would be nice if the authors proofread the manuscript before resubmission.

6. PLOS authors have the option to publish the peer review history of their article (what does this mean?). If published, this will include your full peer review and any attached files.

Reviewer #1: No

Reviewer #2: No

---

## [Author Response · Author response to Decision Letter 0]

16 Apr 2023

Manuscript number: PONE-D-22-18218

Response to reviewer’s comments

Dear Reviewers,

The authors thank you both reviewers for your consideration and valuable comments on the manuscript, title: “Continuous speech with pauses inserted between words increases cortical tracking of speech envelope”, submitting for publication in PLOS ONE. In this document, we have addressed the reviewer’s comments in the following pages. 

Please note that all the reviewer’s comments are in bold text, point-by-point responses to comments are indented normal text. Changes in the manuscript are quoted in “italic text” and indented, line where changes were made are in [red italics].

Best regards,

Suwijak Deoisres

 

Reviewer #1: 

The manuscript reports an EEG experiment investigating the role of added pauses on the cortical tracking of speech. Specifically, as the neural measure for speech tracking, the authors focus on the reconstruction of the speech envelope from the EEG signal (forward/decoding multivariate Temporal Response Function model). The results show that added pauses yield higher envelope reconstruction values compared to natural speech, a finding that seems to be primarily driven by word onsets.

1 - Terminology: entrainment vs tracking

Considering recent debates in the literature (see refs below), I would suggest the authors to use the term “(cortical/neural) tracking of speech” instead of “entrainment”.

Obleser, J., & Kayser, C. (2019). Neural Entrainment and Attentional Selection in the Listening Brain. Trends in Cognitive Sciences, 23(11), 913–926. https://doi.org/10.1016/j.tics.2019.08.004

Meyer, L., Sun, Y., & Martin, A. E. (2020). “Entraining” to speech, generating language? Language, Cognition and Neuroscience, 35(9), 1138–1148. https://doi.org/10.1080/23273798.2020.1827155

R1. Thank you for the suggestion and provided articles. We agree that the terminology “cortical entrainment” should be changed to “cortical tracking”. The terminology was change in the title, abstract, and throughout the main contents.

2 - Goal of the study:

At the beginning of the Introduction, it is mentioned that the goal is to provide the mTRF counterpart of ALRs, while later (lines 137 – 139) the goal shifts to how added pauses modulate the cortical tracking of speech (in terms of forward mTRF). From my understanding, the latter seems to be the main focus throughout the whole manuscript, while the former is barely addressed later on. Could the authors revise this point and provide an explicit and consistent description of their goal(s) in the Introduction? By doing so, the authors could also elaborate more on the theoretical motivations of their goal beyond just its empirical justification.

R2. We have improve the introduction section to be clearer for the reader’s that our main focus is on the effect of added pauses in continuous speech to the cortical envelope tracking, and the mTRF is the tool we used for quantifying the cortical tracking.

A paragraph was added to elaborate more on the theoretical motivation for the study.

“It is widely observed that human brain activity shows tracking of the envelope of stimulus when listening to natural speech [5, 6]. One of the hypothesised functional role of cortical tracking of speech envelope is that it represents the tracking of acoustic onsets [2, 6]. The acoustic onsets in natural speech are generally most commonly linked to the syllable boundaries and regions where speech sounds occur after silent pauses, but it remains unclear how acoustic onsets contribute to cortical envelope tracking. For conventional AERs to repeating stimuli, particularly the auditory late response (ALR), the acoustic onsets can be clearly identified at the start of each sound and it is well established that longer intervals (silent gaps) between stimuli enhances the onset response [7]. A clear observation of strong effects of onsets on cortical responses to natural speech was reported by Hamilton, Edwards [1] using invasive ECoG measurement, in which some regions of the Superior Temporal Gyrus were very sensitive to onset portions of speech, whilst other regions of the Gyrus appeared sensitive to more sustained speech components after onsets. However it is not clear to what extent cortical tracking reflects onset or sustained responses to speech, and hence if we should expect such cortical tracking to increase with the addition of gaps or pauses in continuous speech.” [lines 53-67]

3 - Expected results:

The authors hypothesize stronger cortical responses with added pauses (lines 139-140). However, they do not provide any explanation for this expected result, which might seem even contradictory considering the previous findings reported above (Kayser’s and Hambrook’s studies). Could the authors provide a clear motivation for the mentioned expected results in the Introduction?

R3. We have explain how the stronger cortical responses when pauses were added to speech was hypothesized.

“We hypothesise that the cortical responses would show increase in speech envelope tracking when pauses are added to the stimulus, especially the responses following onsets. This was formulated based on the study by Hamilton, Edwards [1], where the authors reported that cortical response within 200 ms following silent pauses in the speech stream are relatively stronger than later sustained responses. In addition, the continuous speech with additional pauses are becoming more similar to the stimulation of repeating short sound (sound-silence-sound-silence) in ALR measurements, which onset responses may be generated more consistently compared to natural speech. The hypothesis appears to contradict with the previous findings reported by Kayser, Ince [22] and Hambrook, Soni [23], as the two studies showed that irregular speech rhythm (due to change in existed pause durations) and interruption of speech by silent pauses can reduce the listener’s cortical tracking of speech. However, the stimulus manipulation method in the current study is different from the previous studies, the amount of silent pauses in speech increased considerably and speech sounds were not replaced by silent pauses.” [line 108-120]

4 - Rationale of mTRF approach:

Although the authors do specify in the methods section that they will follow the decoding/backward mTRF approach, the terminology used in certain parts of the manuscript is not unambiguous in this respect. In particular, the authors often refer to higher envelope reconstruction correlations as “stronger/enhanced cortical responses”, which might wrongly suggest that encoding/forward mTRF models were used (e.g., referring to higher TRF weights). Moreover, considering that envelope reconstruction correlations could be reflecting phase rather than (only) amplitude alignment between both signals, using terms such as “stronger” or “enhanced” might be misleading. Based on these issues, I suggest the authors (i) specify in the Introduction how mTRF models will be implemented and explain their main rationale (e.g., what does higher envelope reconstruction mean in terms of neural responses?) and (ii) revise how the (interpretation of the) results is conveyed throughout the manuscript.

R4. (i) We specified in the introduction that the decoding/backward mTRF approach will be implemented and the rationale for using the decoding approach is provided in the paragraph below. The interpretation of envelope reconstruction accuracy is also described in the same paragraph. (ii) Thank you for pointing out that the term “stronger/enhance” can be misleading, we have also revised how results is conveyed.

“Two approaches can be employed with the mTRF (Fig 1), either encoding (predicting the EEG using the speech envelope) or decoding (reconstructing the speech envelope from the EEG) to estimate the cortical tracking of speech envelope. While the former follows the causal psychophysiological process of speech driving cortical tracking, the latter has practical advantages in allowing multiple EEG channels to be analysed simultaneously and thus potentially permits more powerful analysis of the association between speech envelope and a set of EEG signals than repeated single channel analyses. Considering these advantages, the decoding approach, will be implemented for data analysis throughout this study. The increase in envelope reconstruction accuracy, quantified by the correlation between the actual and the reconstructed envelope, indicated that the acoustic representation and the neural activity are more synchronous, thus more information may be parsed in the brain for processing of speech sound [7]. Details on the calculation of the backward mTRF (hence will be referred to as the decoder) will be explained in the methods section.” [lines 123-136]

5 - Stimuli:

The method section reports that stimuli were modified by adding short/long pauses between words. Which linguistic (e.g., orthographic/phonological/phonetic?) criteria were used to qualify a speech unit as a word? For example, were contracted forms (e.g., “we’re” instead of “we are”) considered as one or two words? This seems quite relevant considering the use of natural speech, which probably contained many connected speech parts. Providing detailed information about stimuli preparation would be very helpful to better understand the manipulation as well as considering future studies. Also, would it be possible to have access to some of the stimuli?

R5. A sentence has been added to clarify the criteria used to qualify words in the speech stimuli and clarify that contractions are considered as one word.

 “Speech unit were qualified as words based on written spelling (orthographic forms) [27], contractions were considered as one word” [lines 176-177]

We have provided the first segment of the stimuli from the three speech pause conditions, please following the provided link to access the data - https://drive.google.com/drive/folders/1-EDFqFixA3hDVlTfEpkV58UxxfVgPS7T?usp=share_link

Considering the temporal nature of the manipulation, it would be very informative to provide the power spectrum of the stimuli (e.g., Fast Fourier Transform of the speech envelope) as usually done in neural tracking studies. If there were power differences in the speech envelope across conditions, how would they relate to the reported neural tracking findings?

R6. We added a figure (Fig 2) to show the speech envelope modulation spectrum of the natural speech, speech with short pauses, and speech with long pauses stimuli. 

“Fig 2 shows the modulation spectrum of stimuli used across the three speech pause conditions. The peak modulation frequency of the stimuli occurs within the range of approximately 4-5 Hz. The modulation spectrum of stimuli used in the current study appears to be similar to the results reported by Ding, Patel [28].” [lines 183-186]

6 - Behavioural responses:

The results from the behavioural task (comprehension questions) are not included in the manuscript. Could the authors report them? Also, could the authors somehow relate the patterns from the behavioural responses and the envelope reconstruction correlations?

R7. Unfortunately, whilst we asked questions in order to encourage subjects to attend to the stimuli, we did not record the behavioural data. It should also be noted that the responses might not be a good measure to represent behavioural performance as the participants listened to the same story for 3 times in the natural, short, and long pauses condition, so there was some repetition of material. We intended to use the same story across the conditions so that the number of pauses between the short and long pauses condition is the same. The behavioural task was intended to maintain the listener’s attention to the stimulus, so that cortical tracking would not be affected by the level of attention.

7 - Language comprehension/intelligibility:

Although the authors sometimes make some reference to intelligibility issues, it would be interesting to elaborate more on this when explaining the results of the current manuscript. How does the current results relate to the mentioned mixed findings in the literature regarding the relationship between neural tracking and intelligibility? Addressing this important issue would also increase the theoretical implications of the reported results.

R8. As stated above that we do not have the behavioural data to relate with the pattern of cortical tracking of speech envelope, we cannot discuss our current results with the findings in the literature thoroughly. However, we added a paragraph to discuss how our results may be related to the reported findings regarding the relationship between the cortical tracking of speech envelope and intelligibility.

“Considering that the duration of the speech stimulus with additional long pauses expanded more than two times relative to the natural speech stimulus (from approximately 10 minutes to 21 minutes), the intelligibility of the stimulus can change significantly. Although the additional pauses inserted to the stimuli did not alter the speech directly, as there was no time-compression or expansion applied on the temporal waveform, the added pauses may break the phrases or sentences structure and boundary, thus it becomes more difficult to comprehend the ongoing story. Previous studies have shown that the cortical tracking of speech envelope is positively correlated to the behavioural speech-in-noise performance for normal hearing people [4, 36]. Due to the lack of behavioural data in the current study, a clear conclusion on how the increase in cortical tracking when pauses were added to speech relates to individual’s speech comprehension cannot be drawn. However, it is important to consider that the cortical tracking of speech envelope alone may not be an ideal measure to indicate how well a person can understand speech, as it can be influenced by both encoding of acoustic and cognitive processing related to speech comprehension, such as attention to target speech and effort in listening [24, 39]. For example, a listener might show greater cortical tracking of speech envelope when listening to speech in a language they cannot understand than when listening to a language they can understand [39]. Therefore, in this current study, the increase in cortical tracking when pauses were added to speech does not necessarily imply that the participants have improved speech understanding. However, it may be difficult to clearly quantify whether the cortical tracking of speech envelope when pauses were inserted to speech is dominated by the encoding of acoustic information or not.” [lines 444-464]

8 - Onset vs non-onset comparison:

In the methods section “Extraction of speech envelope”, it is said that onsets were taken from the first 150 ms of each word. Because of this approach, relevant linguistic aspects could significantly differ across words, which could somehow impact the reported results. For example, uniqueness point (the specific time point at which that word can be unambiguously identified), informativeness (e.g., how surprising/entropic is that word at that time in terms of phonetic/lexical information?) or syllabic chunking (does that speech segment corresponds to a single syllable or covers partial syllable(s)?) might vary considerably across words. From my perspective, these potential confounds represent limitations for the onset vs non-onset comparison as well as the interpretation of such results.

R9. Thank you for pointing this out. We agree with the reviewer that there is a potential confound between the acoustic onsets/non-onsets and linguistic aspects. However, we think that the inclusion of the analysis on linguistic aspects may expand the scope or content of the current study considerably, as each provided example of linguistic aspects would need to be considered very carefully. We have added a paragraph in the discussion section to include this point as a limitation of our study that it may be an area worth investigating in further studies.

“A limitation of the study was the use of a fixed time interval (150 ms) to represent acoustic onset regions in speech. It should be noted that this time interval does not necessarily correspond with linguistic information. For example the length of syllables varies and the unique time point at which words can be unambiguously identified will vary for different words. As a result the study does have a potential confound between neural process representing acoustic onsets and those representing linguistic boundaries. An interesting area for future work could be to define linguistic boundaries in the speech stream and to compare tracking based on linguistic boundaries to those using fixed acoustic onset regions.” [lines 543-550]

9 - In addition, in lines 402-408, the authors mention a few control analyses/measures for the onset vs non-onset contrast. Could the authors report the results of these control analyses/measures?

R10. We have added Fig 5 to show the result of onset vs non-onset control analysis.

“Fig 6. Amplitude histogram of onset (orange) and non-onset segments (blue) in the delta (top row) and theta frequency bands (bottom row) across the four speech segments.” [lines 393-394] 

Reviewer #2: 

Deoisres et al. explored how inserted pauses between words in natural speech stories modulate neural encoding/decoding of speech signals using a mTRF framework. The results indicate that the encoding/decoding performance is increased by inserted pauses between words.

I found this study interesting. It is interesting to know how the encoding/decoding performance is changed after one inserts pauses between words or slows down normal speech. For hearing-impaired or elderly people, they can process speech better if the speech is slower, which is probably their encoding of word onsets becomes better.

I am positive about this manuscript and here are my comments:

1.The authors do not clearly explain why the study was done. It is difficult to understand the meaning of the study for readers who are not familiar with this field. The interpretation of the data results is not very comprehensive, only the data results are presented, but the interpretation is very limited. More details can be added so that uninitiated readers can find this manuscript more accessible?

R11. We improved the introduction section several points in the introduction to better explain why this study was done. A paragraph was added to point out the gap in knowledge. Please see the response to reviewer#1 comment number 2 “Goal of the study” (R2). 

We have also added a paragraph to discuss more on interpreting the cortical tracking of speech envelope via the envelope reconstruction approach and how it can relate to speech comprehension. Please see the response to reviewer#1 comment number 7 “Language comprehension/intelligibility” (R8).

2. It seems to me that: The authors seem to confuse, sometimes, between the mTRF technique and the continuous speech processing paradigm, and it is felt that the authors did not express the difference between the two clearly. Can the authors make this difference more clear?

R12. We assume that the ‘continuous speech processing paradigm’ is the processing of onsets and non-onsets segmentation from the full speech envelope. We apologise if we misunderstood. A sentence was added to clarify that the processed speech features are input for the mTRF.

“These extracted speech features are acoustic signals assumed to be encoded in the EEG, in which the two signals will then be analysed through the decoder.” [lines 233-235]

We have also revised the wording used through the results and discussions sections to clear up the confusion between the mTRF and speech processing technique.

3.Can the authors spell out ALRs when it first appeared in the manuscript?

R13. ALR is now spelled out when it first appeared. 

“For conventional AERs to repeating stimuli, particularly the auditory late response (ALR).” [lines 58-59]

4.Can the authors elaborate more on how they went from ALRs to mTRF? I understand this logic from my knowledge, but, again, the authors seem to assume that most of readers should understand this, which seems hard just by reading the manuscript.

R14. We elaborate the connection between ALR and cortical tracking of speech envelope, particularly both are related to the acoustic onset effect.

“The acoustic onsets in natural speech are generally most commonly linked to the syllable boundaries and regions where speech sounds occur after silent pauses, but it remains unclear how acoustic onsets contribute to cortical envelope tracking. For conventional AERs to repeating stimuli, particularly the auditory late response (ALR), the acoustic onsets can be clearly identified at the start of each sound and it is well established that longer intervals (silent gaps) between stimuli enhances the onset response [7].” [lines 55-61]

“In addition, the continuous speech with additional pauses are becoming more similar to the stimulation of repeating short sound (sound-silence-sound-silence) in ALR measurements, which onset responses may be generated more consistently compared to natural speech.” [lines 112-115]

5.The total duration for continuous speech stimulus for the no pause, short pause, and long pause conditions were therefore approximately 10 minutes and 20 seconds, 16 minutes, and 21 minutes and 42 seconds, respectively. Does the length difference here affect the encoding/decoding performance? The amount of data sometimes affects values of correlations. Can the authors provide some thoughts on this?

R15. We have provided our thoughts on the effect of stimulus duration on the decoder performance in the discussion section. 

“The longer duration of the stimuli in the short and long pauses conditions may increase the decoding performance due to more training data, however, we have demonstrated in Fig 5 that the increase in stimulus envelope reconstruction accuracy is mainly due to the pause effect. In Fig 5, when the decoder was trained and tested using an equal amount of data across speech pause conditions, the consistent trend of increasing correlation coefficients with longer pauses in speech persists..” [lines 429-434]

6.By the authors' own description in their article, the growth multiplicity of the story has exceeded 0.5 under long pause conditions, which significantly affects the intelligibility of the story, and why the response of mTRF is enhanced under such speech that affects intelligibility, the authors need to explain more on this point.

R16. We have added a discussion on how the increase duration of the stimulus under long pauses conditions might affect the intelligibility of the story. Please see the response to reviewer#1 comment number 7 “Language comprehension/intelligibility” (R8).

7.Line 139. “We hypothesize that the cortical responses to continuous speech would become stronger when pauses are added to the stimulus, especially the responses following onsets. “ I haven’t quite understood this hypothesis. Maybe it is my negligence, but if the authors can make this hypothesis easy to understand, that would help readers in my opinion.

R17. We have explained how we formed the hypothesis. Please see the response on reviewer#1 comment on “Expected results”. 

“We hypothesise that the cortical responses would show increase in speech envelope tracking when pauses are added to the stimulus, especially the responses following onsets. This was formulated based on the study by Hamilton, Edwards [1], where the authors reported that cortical response within 200 ms following silent pauses in the speech stream are relatively stronger than later sustained responses. In addition, the continuous speech with additional pauses are becoming more similar to the stimulation of repeating short sound (sound-silence-sound-silence) in ALR measurements, which onset responses may be generated more consistently compared to natural speech.” [line 108-115]

8.I do not understand the application implications here, and perhaps the authors could provide more detail on the value of comparing the mTRF from different speech corpora.

R18. We have added a paragraph in the discussion section to explain the implications of the current study.

“Future studies incorporating the mTRF paradigm should consider the potential onset effect when comparing measures of cortical tracking obtained from different speech stimuli. The cortical tracking to less natural speech containing more silent pauses, such as the Matrix sentences used for assessing speech reception threshold, may be more influenced by the acoustic onsets [36]. The objective measure values may then become unsuitable for direct comparison and interpretation, as they were estimated using responses and stimuli containing different acoustical properties. For example, the comparison objective measure might not clearly indicate whether one stimulus is more intelligible to the listener than the other. Another practical consideration when using the mTRF paradigm is that it may be suboptimal to apply a linear model trained on one type of speech and testing on a considerably different type of speech. For example, training a model on narrative speech stimulus and testing on Matrix sentences stimulus, and vice versa.” [lines 501-511]

Minor comments

1 The authors lacked a detailed explanation of what mTRF detection time is and the relevant references.

R19. We have changed the phrase “mTRF detection time” to “decoding response using same amount of data across speech pause conditions”, we think this will be clearer to the readers. Rationale for analysis also provided as follow.

“Finally, we examine whether the decoding performance is influenced by the cortical response to speech or simply due to more input data when pauses are added to speech, this is done by limiting the data for analysis to be the same in duration.” [lines 149-151]

2 ‘mTRF was designed to measure the strength of envelope entrainment. ‘

The mTRF does not seem to be specifically designed to measure the strength of envelope entrainment, but has a broader use and more general principles, and if the authors insist on this opinion, they should add relevant references.

R20. We agree that the original statement was not the only intended purpose of the mTRF, so we changed the statement to the following, which describe a more general use of the approach:

“One of the best established tools for measuring human neural response to speech stimuli is the Multivariate Temporal Response Function (mTRF) [6], which is used to quantify the linear relationship between sensory stimulus and its corresponding neural response.” [line 121-123]

3 ‘As it is known that the EEG can entrain to the envelope of speech [7], the mTRF was designed to measure the strength of envelope entrainment. ‘

[7] This article is still based on research under the traditional ERP or isolated stimulus paradigm, and there are many references to mTRF paradigm research.

R21. The previous cited article now changed to [5] Di Liberto, G. M., et al. 2018, which uses the mTRF paradigm, and [6] Howard M.F. and Poeppel D. 2010. 

“It is widely observed that human brain activity shows tracking of the envelope of stimulus when listening to natural speech [5, 6].” [lines 53-54]

4 ‘Randomization was repeated 100 times to construct the null distribution of Pearson's correlation coefficients. ‘

The 100 times of permutation test is a bit low, and it is recommended to try more permutations (500 or 1000) to obtain more stable results.

R22. The permutations were increased to 500 times per each subject’s Pearson’s correlation coefficient. We also updated the result in Fig 5. The main difference from the previous result is the slight increase in number of significant correlation coefficient and a narrower critical band (shaded in grey).

“Randomization was repeated 500 times to construct the null distribution of Pearson’s correlation coefficients.” [lines 293-294]

5 The results of all paired tests might be better presented together with the figures, as two separate presentations may make it difficult for the reader to read and integrate all statistical results. Just a thought.

R23. We have added asterisks in to Fig 4 to indicate pairs with significant difference in Pearson’s correlation. We also add underlines to the headers in Table 1. to indicate which paired speech feature has higher correlation coefficient. We hope that this will improve the presentation of paired comparison results. 

The following sentences were added to the caption of Fig2 and Table 1, respectively.

“Asterisks above paired points indicate significant differences in correlation coefficients (* for p<0.01 and *** for p<0.001).” [Fig 4], [lines 330-331]

“P-values which are underlined indicate that the speech feature labelled at the top of the column with an underline has significantly greater correlation coefficients, or else the other speech feature is greater.” [Table 1], [lines 334-336]

6 There are several typos and some sentences do not sound right. It would be nice if the authors proofread the manuscript before resubmission.

R24. We have proofread the manuscript before resubmission.

---

## [Decision Letter · Decision Letter 1]

31 May 2023

PONE-D-22-18218R1Continuous speech with pauses inserted between words increases cortical tracking of speech envelopePLOS ONE

Dear Dr. Deoisres,

Thank you for submitting your manuscript to PLOS ONE. After careful consideration, we feel that it has merit but does not fully meet PLOS ONE’s publication criteria as it currently stands. Therefore, we invite you to submit a revised version of the manuscript that addresses the points raised during the review process.

We look forward to receiving your revised manuscript.

Kind regards,

Michael Döllinger, Ph.D.

Academic Editor

PLOS ONE

Journal Requirements:

Reviewers' comments:

Reviewer's Responses to Questions

**Comments to the Author**

1. If the authors have adequately addressed your comments raised in a previous round of review and you feel that this manuscript is now acceptable for publication, you may indicate that here to bypass the “Comments to the Author” section, enter your conflict of interest statement in the “Confidential to Editor” section, and submit your "Accept" recommendation.

Reviewer #1: All comments have been addressed

Reviewer #2: All comments have been addressed

2. Is the manuscript technically sound, and do the data support the conclusions?

Reviewer #1: Partly

Reviewer #2: Yes

3. Has the statistical analysis been performed appropriately and rigorously? 

Reviewer #1: Yes

Reviewer #2: Yes

4. Have the authors made all data underlying the findings in their manuscript fully available?

Reviewer #1: No

Reviewer #2: Yes

5. Is the manuscript presented in an intelligible fashion and written in standard English?

Reviewer #1: Yes

Reviewer #2: Yes

6. Review Comments to the Author

Reviewer #1: The authors have addressed my comments from the previous manuscript. However, I still have a few further comments.

- Modulation spectrum of the stimuli (figure 2):

The 3 conditions show similar modulations around the theta-band range (4-8 Hz), but it is also evident that the 2 manipulated conditions show an additional peak in the delta-band range (< 4 Hz) mirroring their respective pause durations (lower frequency peak for long compared to short pauses). Although this is most probably an inevitable consequence of the fixed-duration pause manipulation, currently it remains unclear to what extent these delta-band acoustic modulations might impact the reported results. For example, could this explain why delta-band effects seem generally higher than theta-band effects? I am not necessarily asking for additional control analyses, but I think it is a fair point to be addressed in the manuscript.

- About the hypothesized effect of pauses (only based on Hamilton et al.’s findings), another recent article that could also support this hypothesis partially:

Chalas, Nikos, et al. "Speech onsets and sustained speech contribute differentially to delta and theta speech tracking in auditory cortex." Cerebral Cortex (2023). https://doi.org/10.1093/cercor/bhac502

- Lines 462-464: “However, it may be difficult to clearly quantify whether the cortical tracking of speech envelope when pauses were inserted to speech is dominated by the encoding of acoustic information or not.” I think this affirmation could be rephrased as future work rather than as a problem difficult to solve, considering the possibilities offered by the mTRF approach. In particular, encoding models with different sets of acoustic and/or linguistic regressors are commonly used in the literature to dissociate lower-level (e.g., acoustic) from higher-level (e.g., lexical or semantic components related to comprehension) contributions.

- Lines 561-562: “Cortical responses to natural speech in the delta band appear more strongly linked to non-onset segments, while theta band responses are more related to onset segments.”. This contradicts the lack of significant differences in the natural speech conditions reported in Figure 4.

- Lines 564-567: “The results could be exploited in the clinic in detecting cortical tracking of speech envelope or to onset segments in speech more effectively, where the sensitivity to acoustic onsets could be an indication of how well a person can segment phrases and sentences during speech comprehension.”. I think this somehow contradicts the previous discussion about the uncertainty on the reported findings actually reflecting comprehension-related processes (like syntactic phrasing in this case). Also taking into account the reduced speech naturalness due to the implemented manipulation, what are the clinical implications of this study?

- Minor comment: Figure numbers are missing in the manuscript. Also, figures 3 and 4 show the same results (with and without significance asterisks, respectively).

Reviewer #2: I would like to thank the authors for their time and effort. My concerns have been addressed and I have no further comments. Congratulations!

7. PLOS authors have the option to publish the peer review history of their article (what does this mean?). If published, this will include your full peer review and any attached files.

Reviewer #1: No

Reviewer #2: No

---

## [Author Response · Author response to Decision Letter 1]

15 Jul 2023

Reviewer #1: 

The authors have addressed my comments from the previous manuscript. However, I still have a few further comments.

1 - Modulation spectrum of the stimuli (figure 2):

The 3 conditions show similar modulations around the theta-band range (4-8 Hz), but it is also evident that the 2 manipulated conditions show an additional peak in the delta-band range (< 4 Hz) mirroring their respective pause durations (lower frequency peak for long compared to short pauses). Although this is most probably an inevitable consequence of the fixed-duration pause manipulation, currently it remains unclear to what extent these delta-band acoustic modulations might impact the reported results. For example, could this explain why delta-band effects seem generally higher than theta-band effects? I am not necessarily asking for additional control analyses, but I think it is a fair point to be addressed in the manuscript.

Response 1. We agree that the delta band modulation may strongly affect the cortical tracking in the delta band. We have carried out an additional control analysis to address this potential problem.

We added an additional section with the title “Effect of delta band acoustic modulation on the cortical tracking of speech envelope” [lines 411-412] in the results to report this control analysis, as follows:

(New figure) “Fig 7 caption” [lines 413-416]

 “As shown in Fig 2 that an additional peak in the modulation spectrum appears in the delta band frequency for speech with short and long pause inserted, it is unclear whether the relatively greater correlation coefficients in the delta band compared to the theta band was a result of the delta modulation rate or not. An additional analysis was conducted by removing samples in pause segments and samples in the EEG in lagged time series at the same sample index from the decoding process. This was done to only relate the EEG to the envelope where speech occurs, ensuring a consistent modulation spectrum across different speech pause conditions. This will be referred to as the pauses removed condition. We also included an additional pause removed decoding condition using longer EEG time lag, 0-500 ms, to examine if the original 0-300 ms time lag was sufficient to capture the effect of the delta band modulation rate.

Fig 7 shows the mean correlation coefficients averaged across all participants obtained from decoders trained and test on the full envelope and the envelope with pauses removed in three pauses conditions. Specifically for this analysis, the adjusted significance level for multiple comparisons was adjusted to p≤0.0056 for 9 pairwise comparisons (within each speech pause condition only) in each EEG frequency band. Overall, the same trend in increasing correlation coefficients when longer pauses were inserted to the speech stimulus as reported earlier remains but the correlation values changed significantly when pauses were removed from the decoding process. In the delta band, in the short and long pauses condition, the correlation coefficients when decoding with pauses removed were significantly lower compared to when decoding using the full envelope (p<0.001). While in the theta band, correlation coefficients from decoders with pauses removed were significantly greater than when decoding using the full envelope (p<0.001). Correlation coefficients from pauses removed decoders with different time lags, 0-300 ms and 0-500 ms, were not statistically significant in both the delta and theta band.

The greater correlation coefficients from the decoder in the delta band compared to the theta band appear to be partially affected by the delta band modulation rate in speech with inserted pauses. This is due to the significant decrease in correlation coefficients in the delta band when the pause segments are removed from the decoder analysis, as the EEG delta oscillation may persist in those segments. Despite this, the trend of increasing cortical tracking when pauses are inserted in speech remains. It is also evident that the effect of inserted pauses in speech is stronger for the cortical tracking of speech envelope in the delta band than in the theta band.” [lines 417-446]

We also modified figure 2 by including the modulation spectrum of the stimuli when pauses were removed. [lines 188-192]

2 - About the hypothesized effect of pauses (only based on Hamilton et al.’s findings), another recent article that could also support this hypothesis partially:

Chalas, Nikos, et al. "Speech onsets and sustained speech contribute differentially to delta and theta speech tracking in auditory cortex." Cerebral Cortex (2023). https://doi.org/10.1093/cercor/bhac502

R2. Thank you for sharing the article with us. We agree that the study by Chalas et al. partially supports the hypothesised effect of pauses in speech. We cited the recommended study, in addition to Hamilton et al.’s study. [line 110]

3 - Lines 462-464: “However, it may be difficult to clearly quantify whether the cortical tracking of speech envelope when pauses were inserted to speech is dominated by the encoding of acoustic information or not.” I think this affirmation could be rephrased as future work rather than as a problem difficult to solve, considering the possibilities offered by the mTRF approach. In particular, encoding models with different sets of acoustic and/or linguistic regressors are commonly used in the literature to dissociate lower-level (e.g., acoustic) from higher-level (e.g., lexical or semantic components related to comprehension) contributions.

Response 3. We agree that this should be a possible area for future studies to explore. We have revised the sentences as follows: 

“Future studies may consider implementing the encoding models to dissociate the contribution of lower-level (e.g., acoustic envelope) and higher-level (e.g., phonemes and phonetic features) information of speech to the cortical responses to speech with pauses [5].” [lines 502-504]

4 - Lines 561-562: “Cortical responses to natural speech in the delta band appear more strongly linked to non-onset segments, while theta band responses are more related to onset segments.”. This contradicts the lack of significant differences in the natural speech conditions reported in Figure 4.

Response 4. We edited the sentence to avoid reporting contradictory results. The edited sentences are as follows:

“Cortical responses to natural speech in the delta and theta band are not strongly related to either onset or non-onset segments. However, when pauses were introduced into the speech, responses in both frequency bands become dominated by the onsets.” [lines 601-603]

5 - Lines 564-567: “The results could be exploited in the clinic in detecting cortical tracking of speech envelope or to onset segments in speech more effectively, where the sensitivity to acoustic onsets could be an indication of how well a person can segment phrases and sentences during speech comprehension.”. I think this somehow contradicts the previous discussion about the uncertainty on the reported findings actually reflecting comprehension-related processes (like syntactic phrasing in this case). Also taking into account the reduced speech naturalness due to the implemented manipulation, what are the clinical implications of this study?

Response 5. We agree that these sentences do not align with the previous discussion. It is also not completely clear how the modification of inserting pauses in speech should be exploited in the clinic, given the current findings. Instead, we have therefore changed the sentences to remind the reader that we are uncertain about how the effect of pauses links to speech comprehension, as follows: 

“The influence of the onset responses also led to increased number of cases where cortical tracking of speech envelope was significant, but this may not be an indication of better speech comprehension.” [lines 604-606]

6 - Minor comment: Figure numbers are missing in the manuscript. Also, figures 3 and 4 show the same results (with and without significance asterisks, respectively).

Response 6. We apologise for the incorrect figure that was previously uploaded. The correct figure has been uploaded in its place.

---

## [Editor Report · Decision Letter 2]

17 Jul 2023

Continuous speech with pauses inserted between words increases cortical tracking of speech envelope

PONE-D-22-18218R2

Dear Dr. Deoisres,

We’re pleased to inform you that your manuscript has been judged scientifically suitable for publication and will be formally accepted for publication once it meets all outstanding technical requirements.

Kind regards,

Michael Döllinger, Ph.D.

Academic Editor

PLOS ONE
---

## [Editor Report · Acceptance letter]

20 Jul 2023

PONE-D-22-18218R2 

Continuous speech with pauses inserted between words increases cortical tracking of speech envelope 

Dear Dr. Deoisres:

I'm pleased to inform you that your manuscript has been deemed suitable for publication in PLOS ONE. Congratulations! Your manuscript is now with our production department. 

Kind regards, 

on behalf of

Dr. Michael Döllinger 

Academic Editor

PLOS ONE